# REPLICABLE REINFORCEMENT LEARNING WITH LINEAR FUNCTION APPROXIMATION

**Eric Eaton**
University of Pennsylvania
eeaton@seas.upenn.edu

**Marcel Hussing**
University of Pennsylvania
mhussing@seas.upenn.edu

**Michael Kearns**
University of Pennsylvania
mkearns@cis.upenn.edu

**Aaron Roth**
University of Pennsylvania
aaroth@cis.upenn.edu

**Sikata Sengupta**
University of Pennsylvania
sikata@seas.upenn.edu

**Jessica Sorrell**
Johns Hopkins University
jess@jhu.edu

## ABSTRACT

Replication of experimental results has been a challenge faced by many scientific disciplines, including the field of machine learning. Recent work on the theory of machine learning has formalized replicability as the demand that an algorithm produce identical outcomes when executed twice on different samples from the same distribution. Provably replicable algorithms are especially interesting for reinforcement learning (RL), where algorithms are known to be unstable in practice. While replicable algorithms exist for tabular RL settings, extending these guarantees to more practical function approximation settings has remained an open problem. In this work, we make progress by developing replicable methods for *linear* function approximation in RL. We first introduce two efficient algorithms for replicable random design regression and uncentered covariance estimation, each of independent interest. We then leverage these tools to provide the first provably efficient replicable RL algorithms for linear Markov decision processes in both the generative model and episodic settings. Finally, we evaluate our algorithms experimentally and show how they can inspire more consistent neural policies.

## 1 INTRODUCTION

Replication is a cornerstone of scientific rigor, yet it remains a persistent challenge for the machine learning community (Wagstaff, 2012; Pineau et al., 2021). Especially in reinforcement learning (RL), two runs of the same algorithm on independently sampled traces through a Markov decision process (MDP) may produce dramatically different policies (Henderson et al., 2018). While issues with instability in RL can be traced far back (White & Eldeib, 1994; Mannor et al., 2004), many modern challenges stem from the integration of function approximation techniques such as neural networks into the RL workflow (Islam et al., 2017; Henderson et al., 2018). This instability may be caused by statistical noise (Thrun & Schwartz, 1993), environment perturbations (Pinto et al., 2017), local minima in non-convex optimization landscapes (Bjorck et al., 2022), agents exploring different parts of the state space (Pathak et al., 2017), or the non-stationarity of the data distribution (Baird, 1995; van Hasselt et al., 2018; Voelcker et al., 2025). Even when policies achieve similar average reward, their behavior may be different (Clary et al., 2019; Chan et al., 2020), complicating efforts to verify and build upon existing results. Such inconsistency is particularly problematic in settings where reliability is essential, such as safety-critical or high-stakes applications (García et al., 2015).

To study the limits of stability in learning, a recent line of work (Impagliazzo et al., 2022) introduced a model of replicability in which two executions of the same algorithm must give the exact same output (with high probability). Such guarantees provide a strong benchmark stability and allow us to audit randomized algorithms, as controlling only the algorithm's internal randomness enables exact replication of results. The original paper of Impagliazzo et al. (2022) focused on basic statistical primitives like estimating the value of expectations over a distribution. While this formal notion of replicability is compelling in stationary settings, it requires additional care in settings involving exploration, such as the bandits setting (Esfandiari et al., 2023a). Motivated by the challenge of

producing reliable outcomes in RL (Islam et al., 2017; Henderson et al., 2018; Voelcker et al., 2024), the notion of replicability has recently gained attention in RL research (Eaton et al., 2023; Karbasi et al., 2023). Although exploration poses algorithmic challenges that complicate replicability, many reliability issues in RL may be related to the difficulties of function approximation (Thrun & Schwartz, 1993; van Hasselt et al., 2018). Yet the initial studies on replicability in RL (Eaton et al., 2023; Karbasi et al., 2023) are limited to settings in which one can easily enumerate the state-action space.

In this work, we provide provably replicable methods for linear function approximation in RL. We give the first replicability results for RL beyond the tabular setting: in particular for the *linear* MDP setting (Yang & Wang, 2019; Jin et al., 2020) in which it is assumed that the reward function and transition probabilities are representable as an (unknown) linear function of some common embedding of state-action pairs. This is a setting in which provable learning guarantees are known via function approximation; we give algorithms recovering these provable learning guarantees together with new guarantees of replicability. Our main contributions are as follows:

1. We describe new procedures with first guarantees for (a) replicable regression with random designs and (b) replicable uncentered covariance estimation, both of which may be of independent interest.
2. We apply these tools to develop the first replicable RL algorithms for linear MDPs, encompassing both the generative model and episodic exploration setting.
3. We validate our methods in empirical RL scenarios, demonstrating that they yield replicable or more consistent policies with far fewer samples in practice than required by the theory.

## 2 PRELIMINARIES

We frame the RL problem (Sutton & Barto, 2018) as finding an approximately optimal policy in an episodic MDP (Puterman, 1994) $\mathcal{M} = \{\mathcal{S}, \mathcal{A}, R, P, H, q\}$ with state space $\mathcal{S}$, action space $\mathcal{A}$, reward functions $R = \{R_h\}_{h \in [H]}$, transition kernels $P = \{P_h\}_{h \in [H]}$, horizon $H$, and initial state distribution $q$. In every episode, an agent starts from a state $s_0 \sim q$ and interacts with the MDP for $H$ steps. At any point the agent is in some state $s_h$, chooses an action $a_h$, receives a reward $R_h(s_h, a_h)$, and transitions to a new state according to $P_h(s_{h+1}|s_h, a_h)$. The objective is to find a policy, $\pi = \{\pi_h\}_{h \in [H]}$ that maximizes the expected cumulative reward $V^\pi(q) = \mathbb{E}[\sum_{h=0}^{H-1} r_h(s_h, a_h)]$ where the expectation is over the randomness in the initial state, the policy, and the transitions.

**Notation:** Throughout the text, we will use $\| \cdot \|_p$ to denote the $\ell_p$-norm and if p is omitted $\| \cdot \|$ simply denotes the $\ell_2$ norm; for matrices $\| \cdot \|_F$ denotes the Frobenius norm.

### 2.1 LINEAR MARKOV DECISION PROCESSES

When state-action spaces become large, practitioners often resort to function approximation to solve the RL problem. A common approach to obtaining theoretical guarantees is to make consistency assumptions about the underlying structure of the MDP. We will assume that the rewards and transitions can be represented by a low-dimensional feature representation $\phi : \mathcal{S} \times \mathcal{A} \mapsto \mathbb{R}^d$. This gives rise to the commonly studied framework of linear MDPs (Yang & Wang, 2019; Jin et al., 2020).

**Definition 2.1** (Linear MDP). *$\mathcal{M}$ is a linear MDP with a feature map $\phi : \mathcal{S} \times \mathcal{A} \mapsto \mathbb{R}^d$ if for any $h \in [H]$, there exists $d$ unknown (signed) measures $\mu_h = (\mu_h^1, ... \mu_h^d)$ over $\mathcal{S}$ and an unknown vector $\theta_h \in \mathbb{R}^d$ such that for any state-action pair $(s, a) \in \mathcal{S} \times \mathcal{A}$, we have*

$$R_h(s, a) = \langle \phi(s, a), \theta_h \rangle \quad and \quad P_h(\cdot|s, a) = \langle \phi(s, a), \mu_h \rangle .$$

We make the following structural assumptions on the MDP, that are common in the literature.

**Assumption 2.1** (MDP properties). *All rewards are bounded between $0$ and $1$, the features are normalized such that for all $(s, a)$, $\|\phi(s, a)\| \leq 1$ and for all $h$, we have $\max\{\|\mu(\mathcal{S})\|, \|\theta_h\|\} \leq \sqrt{d}$.*

A fundamental property that makes this framework interesting is that the Q-functions themselves are always contained within the span of the representation, i.e.

**Proposition 2.1** ((Jin et al., 2020)). *For any linear MDP and policy $\pi$, there exists a set of weights $\{\mathbf{w}_h^\pi\}_{h \in [H]}$ such that for all $(s, a, h) \in \mathcal{S} \times \mathcal{A} \times [H]$, we have $Q_h^\pi(s, a) = \langle \phi(s, a), \mathbf{w}_h^\pi \rangle$ .*

Note that simply assuming Proposition 2.1 as our starting point, rather than making the linearity assumptions on the MDP, would not be enough for our algorithms, since we will have to propagate

functions outside the linear class for exploration (Jin et al., 2020). A last fact that will come in useful later is that the weights within each linear MDP can be bounded from above.

**Proposition 2.2** (Jin et al. (2020)). *In a linear MDP, for any fixed policy $\pi$, let $\{\mathbf{w}_h^\pi\}_{h \in [H]}$ be the corresponding weights such that $Q_h^\pi(s,a) = \langle \phi(s,a), \mathbf{w}_h^\pi \rangle$. For all $h$, we have that $\|\mathbf{w}_h^\pi\| \leq 2H\sqrt{d}$ .*

## 2.2 REPLICABILITY

To study RL stability in linear MDPs, we adopt the framework of Impagliazzo et al. (2022), which defines replicability as the demand that a randomized algorithm produce the same output with high probability when run twice with the same internal randomness but independently resampled data.

**Definition 2.2** (Replicability (Impagliazzo et al., 2022)). *Fix a domain $\mathcal{X}$ and target replicability parameter $\rho \in (0,1)$. A randomized algorithm $\mathcal{A}: \mathcal{X}^n \to \mathcal{Y}$ is $\rho$-replicable if for all distributions $D$ over $\mathcal{X}$ and choice of samples $S_1, S_2$, each of size $n$ drawn from $D$, coupled only through the internal randomness $r$ of $\mathcal{A}$, we have: $\boldsymbol{Pr}_{S_1,S_2,r}[\mathcal{A}(S_1;r) \neq \mathcal{A}(S_2;r)] \leq \rho$ .*

Recent work has extended this idea to RL (Eaton et al., 2023; Karbasi et al., 2023), adapting the definition of replicability from the supervised setting; rather than drawing two samples from the same distribution, one asks that two runs of the RL algorithm (with fixed internal randomness) in the same MDP yield identical final policies. For instance, a Q-learning agent with epsilon-greedy exploration interacts with a stochastic environment (external randomness) starting with a randomly initialized policy (internal randomness). Running the agent twice on the environment with the same internal random seed should produce the same policy with high probability. Obtaining identical policies, rather than merely achieving similar rewards, is crucial for predictable and analyzable behavior. This stricter requirement eliminates subtle but potentially impactful behavioral differences that arise from variations in the training data and cause drastic failure in safety-critical domains (García et al., 2015).

Our results rely on a key technique by Impagliazzo et al. (2022) that uses randomized rounding to obtain replicability in vector spaces, which we extend to matrix spaces. We state a slightly adapted version of their results in Algorithm 1 and Theorem 2.1.

---

**Algorithm 1** `R-Hypergrid-Rounding` (adapted from Algorithm 6 in (Impagliazzo et al., 2022))

---

**Input:** Matrix $A \in \mathbb{R}^{d_1 \times d_2}$ with entries bounded between $b_\mathrm{s}$ and $b_\mathrm{e}$, shared randomness $r$, rounding accuracy $\alpha$

---

1: Uniformly at random draw $\alpha^\mathrm{off}$ from $[0, \alpha)^{d_1 \times d_2}$ using $r$
2: $\forall i \in [d_1], j \in [d_2]$, define the set of grid intervals
$\{[\, b_\mathrm{s},\ b_\mathrm{s}+\alpha_{i,j}^\mathrm{off}), [\, b_\mathrm{s}+\alpha_{i,j}^\mathrm{off},\ b_\mathrm{s}+\alpha_{i,j}^\mathrm{off}+\alpha), [\, b_\mathrm{s}+\alpha_{i,j}^\mathrm{off}+\alpha,\ b_\mathrm{s}+\alpha_{i,j}^\mathrm{off}+2\alpha), \ldots, [\, b_\mathrm{s}+\alpha_{i,j}^\mathrm{off}+\kappa\alpha,\ b_\mathrm{e}\,)\}$,
3: Form $\bar{A}$ by mapping each entry $A_{i,j}$ to the midpoint of the (unique) grid interval from the above set that contains $A_{i,j}$.
4: **Return** $\bar{A}$.

---

Algorithm 1 will return a version of the input matrix $A$ that has been rounded to a randomly shifted grid of intervals in each dimension. This rounded version is a replicable estimate that does not differ too much from the original estimate, as formalized in the following:

**Lemma 2.1** (adapted from (Impagliazzo et al., 2022)). *Let $\mathcal{A}$ be Algorithm 1. Let $A^{(1)}, A^{(2)} \in \mathbb{R}^{d_1 \times d_2}$ with entries bounded between $b_s$ and $b_e$ where the bounds are solely required for computational purposes. For both matrices $A^{(a)}$, $a \in [1,2]$, we have $\|A^{(a)} - \mathcal{A}(A^{(a)})\|_F \leq \sqrt{d_1 d_2}\frac{\alpha}{2}$. Further, denote $\|A^{(1)} - A^{(2)}\|_F = \Delta$. Then, $\boldsymbol{Pr}[\mathcal{A}(A^{(1)}) = \mathcal{A}(A^{(2)})] \geq 1 - d_1 d_2 \frac{\Delta}{\alpha}$ .*

*Proof.* If the Frobenius norm between the two matrices $A^{(1)}, A^{(2)}$ is at most $\Delta$, then by a Frobenius to $\ell_1$ conversion the $(i,j)$th coordinate of the two matrices is not rounded to the same point with probability $\frac{|A_{i,j}^{(1)} - A_{i,j}^{(2)}|}{\alpha}$. A union bound over the matrix size proves the claim about replicability. For accuracy, the algorithm will change each element by at most $\alpha/2$, resulting in an $\ell_1$ bound on the matrix difference. Converting from the elementwise difference to Frobenius completes the proof. $\square$

## 3 REPLICABLE TOOLS FOR LINEAR SPACES

Before discussing replicable RL, we first establish two algorithms that are useful for working with data under linearity assumptions and that will be crucial components of our RL procedures. These include a *replicable linear regression estimator* as well as a *replicable second order moment estimation* procedure. Given the widespread use of linear estimators across scientific disciplines, we believe that these methods hold broader methodological relevance beyond their use for replicable RL.

This section will first consider a supervised setting where we are given a dataset of input variables $\mathbf{x} \in X \subseteq \mathbb{R}^d$ and corresponding labels $y \in \mathbb{R}$. The dataset is drawn independently from some distribution $D$, and the data is modeled using a linear function $f(\mathbf{x}) = \mathbf{w}^\top \mathbf{x}$, where $\mathbf{w} \in \mathbb{R}^d$ is the vector of model weights. We will make the following assumption about boundedness that 1) ensure the resulting optimization problems remain well-defined and stable and 2) prevent extreme values from disproportionately influencing the learned models

**Assumption 3.1** (Boundedness of Inputs, Labels, and Weights). *The input vectors $\mathbf{x} \in \mathbb{R}^d$ satisfy $\|\mathbf{x}\| \leq 1$, the labels $y \in \mathbb{R}$ satisfy $|y| \leq Y$, and all model weights $\mathbf{w} \in \mathbb{R}^d$ satisfy $\|\mathbf{w}\| \leq B$ .*

### 3.1 REPLICABLE RIDGE REGRESSION

To effectively operate within the linear MDP space, a first step is to develop a procedure for replicable linear regression. Prior work on bandits provided a replicable least-squares estimator in the fixed design setting (Esfandiari et al., 2023a) where one has access to a distribution $\nu$ of a fixed set input vectors $\mathbf{x}$. This approach is limited to scenarios in which a fixed design distribution can be obtained replicably. In addition, it assumes that the design distribution sufficiently spans the input space. Both of these conditions are not common in RL, where the distribution of visited states evolves as the agent explores its environment. Our first contribution is a novel algorithm that builds on the well-known ridge regression algorithm (Hoerl & Kennard, 1970) to circumvent these issues. This algorithm works both in scenarios where the full space is not spanned as well as in problems with random design.

Our `R-Ridge-Regression` algorithm is given in Algorithm 2 and applies replicable rounding as described in Algorithm 1 to the weights output by classical ridge regression. The key insight that allows us to ensure replicability is that the ridge regressor converges to a global optimum due to the strong convexity of the minimizer. While many results in replicability rely on closeness to a ground truth parameter, we simply require that the algorithm can minimize the given objective. This lets us relate approximations in objective to approximation in parameter space. Note that we are not making any assumption about the data we are given at this point and the results hold even in the agnostic case. The following theorem quantifies the amount of data needed to obtain a replicable estimate.

**Theorem 3.1.** *Suppose Assumption 3.1 holds. Let $\varepsilon, \delta, \rho \in (0, 1)$. Let $D_{[t]} = \{D_1, ..., D_t\}$ be a sequence of independent distributions. Let $S \sim D_{[t]}^M$ denote a sample generated by taking $M$ i.i.d. draws from each of the $t$ distributions in $D_{[t]}$. Denote $N = tM$. For any $D_{[t]}$ we have that Algorithm 2 is $\rho$-replicable. Let $\widetilde{\theta} = \arg\min_\theta \mathbb{E}_{D_{[t]}^M}[(\theta^\top \mathbf{x} - y)^2 + \lambda\|\theta\|_2^2]$. Then with probability at least $1 - \delta$ over choice of the sample $S \sim D_{[t]}^M$, it holds that $\|\overline{\mathbf{w}} - \widetilde{\theta}\| \leq \varepsilon$ as long as the number of samples drawn is $N \in \Omega\left(\frac{(B+Y)^2 d^3}{\lambda^2 \varepsilon^2 (\rho - 2\delta)^2} \log\left(\frac{1}{\delta}\right)\right)$ .*

As mentioned before, the proof for this result uses the fact that ridge regression provides a strongly convex objective. Using this observation, one could first derive a standard excess-risk bound for ridge regression using classical real-valued uniform convergence tools such as Rademacher complexity,

---

**Algorithm 2** `R-Ridge-Regression`

    **Input:** Data $\mathcal{D} = \{\mathbf{x}_i, y_i\}_{i=1}^N$, regularization parameter $\lambda$, accuracy $\varepsilon$, failure probability $\delta$, replicability parameter $\rho$, shared random string $r$

1: Compute $\widehat{\mathbf{w}} = (\sum_{i=1}^N \mathbf{x}_i \mathbf{x}_i^T + \lambda I)^{-1} \sum_{j=1}^N \mathbf{x_j} y_j$
2: $\overline{\mathbf{w}} = $ `R-Hypergrid-Rounding`$(\widehat{\mathbf{w}}, r, \frac{d\varepsilon}{d^{3/2} + \rho - 2\delta})$
3: **return** $\overline{\mathbf{w}}$

---

and then invoke strong convexity to translate an excess objective tolerance $\varepsilon' > 0$ into a parameter-distance bound of the form $\|\hat{\mathbf{w}} - \widetilde{\theta}\| \in O\big(\sqrt{\varepsilon'/\lambda}\big)$, where $\hat{\mathbf{w}}$ is the ridge solution from Algorithm 2 and $\lambda$ is the regularization parameter. Yet, in our replicability setting, the rounding step requires the weights produced in two independent executions to be extremely close in Euclidean norm; achieving this level of parameter closeness via such a risk-based argument would require an extremely small excess-risk tolerance, and feeding this tolerance back into the uniform convergence bound leads to a substantially worse polynomial dependence on $d$ than the rate established in our theorem.

Instead, we rely on a stronger guarantee. Specifically, we show if the population gradient of the risk is uniformly close to the empirical gradient, that is if $\sup_\theta |\nabla_\theta R(\theta) - \nabla_\theta \widehat{R}(\theta)| \leq \varepsilon'$, then strong convexity implies that the distance between the empirical and true minimizer can be bounded as $\|\hat{\mathbf{w}} - \widetilde{\theta}\| \in O(\varepsilon'/\lambda)$ (without square root). We formalize this in Lemma A.2. It remains to prove that this gradient difference is in fact small. A traditional uniform convergence analysis is insufficient as the gradient is vector-valued. Thus, to get the bound in our theorem, we leverage a recent result by Foster et al. (2018) that provides efficient gradient uniform convergence based on Rademacher results in (Bartlett et al., 2005) and a vector-valued symmetrization lemma (Maurer, 2016). These results are largely for independent and identically distributed data. To obtain our multi-distributional result, we generalize the Rademacher results by Bartlett et al. (2005) to multiple distributions in Appendix A.1 and propagate this change through all relevant results. Then, we prove in Theorem A.2 that the ridge empirical gradient is close to the population gradient. The key feature of this analysis is that it comes with no dependence on $d$ just like the traditional risk analysis even though we are analyzing a $d$-dimensional object. This gives us all tools to conclude a replicability result in Theorem 3.1. Now, we argue that replicability can be achieved via rounding the weights by relying on the parameter closeness established by strong convexity. We provide the full proof in Appendix A.

Theorem 3.1 only establishes replicability; the accuracy of the algorithm is determined by the shape of the underlying data distribution. To get accuracy guarantees, we will make the standard assumptions that the underlying functional structure of the labels is in fact linear and the labels have $0$ mean. Furthermore, we will assume access to a distribution over a core set of vectors that allows us to represent every point on the domain. More precisely, we define the following core set.

**Definition 3.1** (Core set). *Let $\nu(\mathbf{x}_i)$ be a design distribution over vectors $\mathbf{x}_i \in C_k \subseteq \mathbb{R}^d$. We call $C_k = \{\mathbf{x}_i\}_{i=1}^k$ a core set if it satisfies that every vector in the domain $X$ can be written as a linear combination of points on the support of $\nu$, i.e. $\mathbf{x} = \sum_{\mathbf{x}_i \in \mathrm{supp}(\nu)} \eta_i \nu(\mathbf{x}_i) \mathbf{x}_i$ with $\|\eta\|_2^2 \leq k$ .*

Note that we are not making any second order moment assumptions in our core set definition, which means standard least squares estimation would not be possible. Yet, given such a core set, we can give a direct bound on the prediction error of the `R-Ridge-Regression` procedure.

**Theorem 3.2** (Fixed Design R-Ridge Regression Error). *Suppose Assumption 3.1 holds. Consider a dataset $\mathcal{D} = \{\mathbf{x}_i, y_i\}_{i=1}^N$ where $\mathbf{x}_i$ comes from a coreset $C_k$. Let $\mathbf{x}$ be drawn i.i.d. from $C_k$'s distribution $\nu(\mathbf{x})$. For each $y_i$, let $y_i = (\theta^*)^T \mathbf{x}_i + \epsilon_i$ where $\{\epsilon_i\}_{i=1}^N$ are independent random variables with $\mathbb{E}[\epsilon_i] = 0$. Let $\varepsilon, \delta, \rho \in (0, 1)$. As long as we draw $N \in \Omega\left(\frac{(B+Y)^2 d^3 k^2 \|\theta^*\|^4}{\varepsilon^6 (\rho - 2\delta)^2} \log\left(\frac{1}{\delta}\right)\right)$ samples, Algorithm 2 is $\rho$-replicable, and with probability $1 - \delta$ it holds that $\max_x |x^\top (\overline{\mathbf{w}} - \theta^*)| \leq \varepsilon$ .*

To prove the accuracy bound (see Appendix A.4, we use our earlier analysis to control the distance between the output $\overline{\mathbf{w}}$ of Algorithm 2 and the ridge minimizer $\widetilde{\theta}$. By the triangle inequality, $\max_{\mathbf{x}} |\mathbf{x}^\top (\overline{\mathbf{w}} - \theta^*)|$ is then bounded by the sum of $\max_{\mathbf{x}} |\mathbf{x}^\top (\overline{\mathbf{w}} - \widetilde{\theta})|$ and $\max_{\mathbf{x}} |\mathbf{x}^\top (\widetilde{\theta} - \theta^*)|$, so it remains to control the second term. The coreset assumption implies that every $\mathbf{x}$ in the support of $\nu$ can be written as a linear combination of at most $k$ coreset points with a coefficient vector of norm at most $\sqrt{k}$, which yields the bound $|\mathbf{x}^\top (\widetilde{\theta} - \theta^*)| \leq \sqrt{k} \big(\mathbb{E}_{\mathbf{x} \sim \nu}[(\mathbf{x}^\top (\widetilde{\theta} - \theta^*))^2]\big)^{1/2}$. Using the optimality of the ridge solution $\widetilde{\theta}$ for the model $y = \theta^\top \mathbf{x} + \epsilon$, we show that $\mathbb{E}_{\mathbf{x} \sim \nu}[(\mathbf{x}^\top (\widetilde{\theta} - \theta^*))^2] \leq \lambda |\theta^*|^2$, which implies the uniform bound $|\mathbf{x}^\top (\widetilde{\theta} - \theta^*)| \leq \sqrt{k\lambda} |\theta^*|$ for all $\mathbf{x}$. Choosing $\lambda = \varepsilon^2 / (4k |\theta^*|^2)$ makes this term at most $\varepsilon/2$; combining it with the $\varepsilon/2$ bound on $|\mathbf{x}^\top (\overline{\mathbf{w}} - \widetilde{\theta})|$ yields the guarantee.

We acknowledge that the generality leads to slightly worse bounds than those achieved in the fixed design setting. This is because we require a technique that works even when the data support is small. If one is instead able to make assumptions about the eigenvalues of the second order moment one can recover tighter rates, close to the fixed design rates of (Esfandiari et al., 2023a) by setting $\lambda = 0$.

---

**Algorithm 3** `R-UC-Cov-Estimation`

---

**Input:** Data $\mathcal{D} = \{\mathbf{x}_{i,m}\}_{(i,m)\in[t]\times[M]}$, accuracy $\varepsilon$, failure probability $\delta$, replicability parameter $\rho$, shared random string $r$

---

1: Compute $\widehat{\Sigma}_{jl} = \sum_{i=0}^{t-1} \frac{1}{M} \sum_{m=0}^{M-1} \mathbf{x}_{i,m}^j \mathbf{x}_{i,m}^l, \quad 1 \leq j \leq l \leq d.$

2: $\overline{\Sigma}_{jl} \leftarrow$ `R-Hypergrid-Rounding`$\left(\widehat{\Sigma}_{jl}, r, \frac{d^2\varepsilon}{(d^3+\rho-2\delta)}\right)$

3: $\overline{\Sigma}_{lj} \leftarrow \overline{\Sigma}_{jl}$ for all $l < j$ $\hspace{5cm}$ ▷ symmetrize

4: Return $\Pi_{\mathrm{PSD}}(\overline{\Sigma})$ $\hspace{2cm}$ ▷ $\Pi_{\mathrm{PSD}}(A) = U \, \mathrm{diag}(\max(\zeta, 0)) \, U^\top$ for $A = U \, \mathrm{diag}(\zeta) \, U^\top$

---

## 3.2 REPLICABLE UNCENTERED COVARIANCE ESTIMATION

The second tool we need is a procedure for obtaining a replicable estimate of the second order moment matrix, which we will use to identify parts of the state space that have been visited. In Algorithm 3 we provide this replicable estimation procedure. Two features of a covariance matrix are that it is symmetric and positive semidefinite (PSD). Simply applying element-wise randomized rounding to the regular covariance matrix might lead to an output matrix that is neither symmetric nor PSD. Thus, our algorithm first computes the upper-triangular part of the regular uncentered covariance, randomly rounds it, and then symmetrizes explicitly. Finally, we project the matrix back onto the original cone by clipping its eigenvalues to ensure the algorithm's output is PSD. We guarantee replicability and closeness in Frobenius norm to the expected uncentered covariance via the following Theorem.

**Theorem 3.3.** *Suppose Assumption 3.1 holds. Let $\varepsilon, \delta, \rho \in (0, 1)$. Let $D_{[t]} = \{D_1, ..., D_t\}$ be a sequence of independent distributions. Let $S \sim D_{[t]}^M$ denote a sample generated by taking $M$ i.i.d. draws from each of the $t$ distributions in $D_{[t]}$. Denote $N = tM$. Algorithm 3 is $\rho$-replicable. With probability at least $1 - \delta$ over the independent draw of the dataset $\mathcal{D} = \{\mathbf{x}_{t,m}\}_{(t,m)\in[T]\times[M]}$, it holds that $\|\Pi_{\mathrm{PSD}}(\overline{\Sigma}) - \mathbb{E}[\mathbf{x}\mathbf{x}^\top]\|_F \leq \varepsilon$ as long as we draw $N \in \Omega\left(\frac{d^8 t^2}{\varepsilon^2(\rho-\delta)^2} \log\left(\frac{d^2}{\delta}\right)\right)$ samples.*

To prove the theorem, we apply an element-wise concentration inequality to the upper-triangular part of the empirical covariance matrix and then symmetrize it by copying these entries to the lower-triangular half. The rounding procedure may cause some of the smaller eigenvalues of this matrix to become negative, so we project back onto the cone by clipping these eigenvalues. Since the true uncentered covariance matrix lies in this cone, this projection cannot increase the estimation error. Moreover, as $|x| \leq 1$ for all $x \in \mathcal{X}$, the covariance matrix is uniformly bounded. Combining these observations with Lemma 2.1 yields the stated result. The full proof is provided in Appendix B.1.

## 4 REPLICABLE RL WITH LINEAR FUNCTION APPROXIMATION

Equipped with the tools to handle linear spaces replicably, we now present our main results: replicable algorithms for linear MDPs in the generative model and the more challenging episodic setting.

### 4.1 REPLICABLE LINEAR RL WITH GENERATIVE MODELS

As a warmup, we will consider the setting of RL with a generative model (Kearns & Singh, 1998). In this setting, one is given access to a generative model $G_{\mathcal{M}}$ that given a state $s_h$ and an action $a_h$ returns a deterministic reward $R_h$ and a next state $s_{h+1}$ sampled from the transition probability $P_h(\cdot|s_h, a_h)$. In the tabular setting, it is common to simply sample every state-action pair from the environment sufficiently often until enough data for statistical concentration is available (e.g. (Kearns & Singh, 1998; Kakade, 2003)). In the linear setting this is unfortunately not possible since there are possibly infinitely many state-action pairs. Instead, we need to obtain a set of *representative* state-action pairs that will cover the lower dimensional space of $\phi$. Such a set of vectors is often assumed given and can be represented via a set of states that gives Mahalanobis distance guarantees (Yang & Wang, 2019) or via an optimal design (Lattimore & Szepesvári, 2020). Given that `R-Ridge-Regression` works without second order moment assumptions, we will reuse our core set as given in Definition 3.1.

We state our algorithm for replicable RL with a generative model and access to a core set in Algorithm 4. Intuitively, the algorithm produces an i.i.d. dataset of size $M$ for every $h$ by drawing

---

**Algorithm 4** `R-LSVI` with core set

---

**Input:** MDP $\mathcal{M}$, state action pairs (for core set) $C$, accuracy $\varepsilon$ and failure probability $\delta$, replicability parameter $\rho$, random string $r$

---

1: $M \leftarrow \Omega\left(\frac{d^6 k^3 H^{22}}{\varepsilon^8 (\rho - 2\delta)^2} \log\left(\frac{H}{\delta}\right)\right)$; $\lambda \leftarrow \Omega(\frac{\varepsilon^2}{kH^2 d})$
2: $\hat{V}_{H+1}(\cdot) = \vec{0}$
3: **for** $h = H$ to $1$ **do**
4: $\quad \mathcal{D} = \{\phi(s,a), R_h(s,a) + \hat{V}_{h+1}(s')\}_{(s,a) \in C, s' \sim G_{\mathcal{M}}^{\nu(s,a) M}(s,a)}$
5: $\quad \hat{\mathbf{w}}_h^\top = $ `R-Ridge-Regression`$\left(\mathcal{D}, \lambda, \frac{\varepsilon}{2H^2}, \frac{\delta}{H}, \frac{\rho}{H}, r\right)$
6: $\quad \hat{Q}_h(\cdot) = \hat{\mathbf{w}}_h^\top \phi(\cdot)$
7: $\quad \hat{V}_h(\cdot) = \min\{\max_a \hat{Q}_h(\cdot, a), H\}$
8: **end for**
9: **return** $\{\hat{\pi}_h(s)\}_{h=1}^H$, s.t. for all $h$, $\hat{\pi}_h(s) = \arg\max_a \hat{Q}_h(s,a)$

---

next states from the generative model according to the distribution of the core set. It then computes the value of the current from the next time-step iteratively. As we use a replicable estimation procedure to obtain the weights $\mathbf{w}_h$ at each round, we are guaranteed that estimates in every run will be replicable as long as we draw sufficiently many samples. This is formalized in the following statement.

**Theorem 4.1** (Sample Complexity `R-LSVI` with core set). *Let $\mathcal{M}$ be a linear MDP and suppose Assumption 2.1 holds. Suppose we have access to a set of state action pairs $C$, s.t. the corresponding vectors $\phi(s,a)$ form a core set $C_k$ of the lower dimensional space of $\phi$. Let $\delta, \varepsilon, \rho \in [0,1]$. Algorithm 4 is $\rho$-replicable and with probability $1 - \delta$ outputs a list of policies $\{\hat{\pi}_h\}_{h=1}^H$ that guarantees us*
$$\forall s \in \mathcal{S}, \quad |V^*(s) - V^{\hat{\pi}}(s)| \leq \varepsilon \text{ as long as we draw } N \in \Omega\left(\frac{d^6 k^3 H^{23}}{\varepsilon^8 (\rho - 2\delta)^2} \log\left(\frac{H}{\delta}\right)\right) \text{ samples.}$$

The accuracy proof combines the standard core set ideas (Lattimore & Szepesvári, 2020) with Theorem 3.2, which ensures that each stage's ridge regression produces Q-estimates with Bellman error at most $\varepsilon'$. A standard dynamic-programming argument then shows that for all $s$ we have $|V^*(s) - V^{\hat{\pi}}(s)| \leq 2H^2 \varepsilon'$. Choosing $\varepsilon'$ appropriately yields the claimed accuracy. For replicability, we know the only randomness comes from sampling. We start with a fixed initialization; then proof by induction that the value estimates remain replicable. The full proof is provided in Appendix C.

## 4.2 REPLICABLE LINEAR RL WITH EXPLORATION

The previous section illustrated that it is possible to obtain replicable algorithms in the linear MDP setting. However, so far we assumed that we have access to a specific core set of state-action pairs of which we can draw next-state samples as we please. A key challenge for replicability is to deal with the noisy exploration process in RL. In the exploration setting, it is possible to obtain optimal policies that are non-replicable using LSVI-UCB (Jin et al., 2020). This section builds on this well-established finding and Algorithm 5 provides a replicable version of LSVI-UCB called `R-LSVI-UCB`.

`R-LSVI-UCB` proceeds in rounds. Rather than updating the policy at every episode, it collects a batch of sampled data with the current policy to obtain replicable estimates of the required data-dependent quantities. The algorithm then uses `R-Ridge-Regression` to obtain a mapping from $\phi$ to a prediction of $Q$. For exploration, we add a upper confidence bound (UCB) bonus term computed via `R-UC-Cov-Estimation`. This leads to the following guarantee for `R-LSVI-UCB`.

**Theorem 4.2** (Sample Complexity `R-LSVI-UCB`). *Let $\mathcal{M}$ be an episodic linear MDP and suppose Assumption 2.1 holds. Let $\varepsilon, \delta, \rho \in (0,1)$. Algorithm 5 is $\rho$-replicable and after collecting a total of $MT \in \Omega\left(\frac{d^{56} H^{62} \log^5(1/\delta)}{\varepsilon^{44} \rho^2}\right)$ trajectories, and outputs a list of policies $\Pi^T = \{\hat{\pi}^t\}_{t=0}^T$ such that with probability $1 - \delta$, for all $\pi \in \Pi$, $\mathbb{E}_{\pi^t \sim \Pi^T, s_0 \sim q}[V^\pi(s_0) - V^{\pi^t}(s_0)] \leq \varepsilon$ .*

The full proof is provided in Appendix D. While our analysis resembles the high-level ideas of LSVI-UCB by Jin et al. (2020), additional ingredients are needed to prove both the replicability and accuracy guarantees in Theorem 4.2. We cannot simply plug our primitives for linear spaces into the fully online RL loop, as they only work for fixed-size batches drawn from a fixed distribution. Instead, we analyze a batched variant of LSVI-UCB in which the policy is held fixed within each round and

---

**Algorithm 5** `R-LSVI-UCB`

---

**Input:** MDP $\mathcal{M}$, accuracy $\varepsilon$, failure probability $\delta$, replicability parameter $\rho$, random string $r$

---

1: $T \leftarrow \tilde{\Omega}\left(\frac{\beta^2 H^2 d \log(1/\delta)}{\lambda \varepsilon^2}\right)$; $M \leftarrow \tilde{\Omega}\left(\frac{T d^8 \log 1/\delta}{\Delta_\Lambda^2 \rho_{est}^2}\right)$; $\beta \leftarrow \tilde{\Omega}(dH)$;

2: $\lambda \leftarrow \Omega(\frac{\varepsilon^2}{H^2 d^2})$; $\Delta_w \leftarrow O(\frac{\varepsilon}{H})$; $\Delta_\Lambda \leftarrow O(\lambda^5 (\frac{\varepsilon}{\beta H})^4)$; $\rho_{\text{est}} = \Omega(\frac{\rho}{TH})$; $\delta_{\text{est}} = \Omega(\frac{\delta}{TH})$;

3: $\hat{Q}_h^0(\cdot,\cdot) = \lambda\beta\|\phi(\cdot,\cdot)\|$ for all $h \in [H]$

4: $\hat{V}_h^0(\cdot) = \max_{a\in\mathcal{A}} \hat{Q}_h^0(\cdot,a)$ for all $h \in [H]$

5: $\hat{\pi}^0 = \{\hat{\pi}_h^0\}_{h\in[H]}$ where $\hat{\pi}_h^0(s) = \arg\max_{a\in\mathcal{A}} \hat{Q}_h^0(s,a)$

6: **for** round $t \in [T]$ **do**

7:     **for** $m \in [M]$ **do**                                    ▷ Sample $M$ trajectories under new policy

8:         Observe starting state $s_{m,0}^t \sim q$

9:         **for** $h \in [H]$ **do**

10:             Take action $a_{m,h}^t \leftarrow \hat{\pi}^t(s_{m,h}^t)$ and receive reward $R_{m,h}^t$

11:         **end for**

12:     **end for**

13:     **for** $h \in [H]$ **do**

14:         **for** $m \in [M]$ **do**

15:             $\mathbf{x}_{m,h}^t = \phi(s_{m,h}^t, a_{m,h}^t)$

16:             **for** $i \in [t]$ **do**

17:                 $y_{m,h}^i = R_{m,h}^t + \hat{V}_{h+1}^t(\mathbf{x}_{m,h+1}^t)$

18:             **end for**

19:         **end for**

20:         $\bar{\mathbf{w}}_h^{t+1} \leftarrow$ `R-Ridge-Regression`$(\{(\mathbf{x}_{m,h}^i, y_{m,h}^i)\}_{m\in[M], i\in[t]}, \lambda, \Delta_w, \delta_{\text{est}}, \rho_{\text{est}}, r)$

21:         $\bar{G}_h^{t+1} \leftarrow$ `R-UC-Cov-Estimation`$(\{\mathbf{x}_{m,h}^i\}_{m\in[M], i\in[t]}, \Delta_\Lambda, \delta_{\text{est}}, \rho_{\text{est}}, r)$

22:         $\bar{\Lambda}_h^{t+1} \leftarrow \bar{G}_h^{t+1} + \lambda I$

23:         $\hat{Q}_h^{t+1}(\cdot,\cdot) = \min\{H, \langle\bar{\mathbf{w}}_h^{t+1}, \phi(\cdot,\cdot)\rangle + \beta[\phi(\cdot,\cdot)^T(\bar{\Lambda}_h^{t+1})^{-1}\phi(\cdot,\cdot)]^{1/2}\}$

24:         $\hat{V}_h^{t+1}(\cdot) = \max_{a\in\mathcal{A}} \hat{Q}_h^{t+1}(\cdot,a)$

25:     **end for**

26:     $\hat{\pi}^{t+1} = \{\hat{\pi}_h^{t+1}\}_{h\in[H]}$ where $\hat{\pi}_h^{t+1}(\cdot) = \arg\max_{a\in\mathcal{A}} \hat{Q}_h^{t+1}(\cdot,a)$

27: **end for**

28: Return $\{\hat{\pi}^t\}_{t\in[T]}$

---

a fresh batch of trajectories is collected. This batching breaks a key algorithmic symmetry in the original LSVI-UCB analysis, where both regression and the bonus use the same Gram matrix over all past data. As a result, Theorem 4.2 requires a new regret argument for this perturbed-Gram setting (Lemma D.5, Corollary D.1) together with a new inter-policy value difference bound (Lemma D.1).

The optimism and regret guarantees in LSVI-UCB rely on computing the bonus from the empirical regularized Gram matrix and applying an elliptical potential argument. Under rounding, this argument no longer applies directly. We develop a novel perturbation analysis that bounds Mahalanobis norms under positive semidefinite perturbations of the Gram matrix (Lemmas D.2 and D.3). This shows that the rounded-covariance bonus still yields an optimistic Q-function that drives exploration (Lemma D.4) and that its cumulative contribution to regret matches the unrounded rate.

The guarantees for the replicable ridge and uncentered covariance routines only apply when each call receives i.i.d. samples from a fixed distribution. In the adaptive, episodic setting the data distributions in later rounds depend on earlier policies. To reconcile this, we prove replicability by strong induction over rounds: conditioning on the event that both executions have produced identical policies up to round $t$, the data in round $t + 1$ is a draw from a fixed distribution determined by the shared internal randomness and hence identical across runs. We refer to Appendix D.2 for more details.

## 4.3 LIMITATIONS

The two algorithms we present both give strong stability guarantees in a linear function approximation MDP setting. However, their sample complexity cost is larger than that of their non-replicable

counterparts; additionally, linear MDPs alone are often insufficient in practice. While recent work has shown promising results employing linear MDPs on common benchmarks Zhang et al. (2022), they require a meticulous feature learning procedure that has no replicability guarantees. Towards fully practical replicability, the feature learning problem for low-rank MDPs (Jiang et al., 2017; Du et al., 2020; Agarwal et al., 2020; Modi et al., 2024) remains an interesting open question.

## 5 EXPERIMENTAL EVALUATION

While some of our worst-case guarantees might seem impractical, this section shows that in practice, our algorithms need far fewer samples to work effectively. We also show that even though our results are largely derived for linear MDPs with fixed feature representations, the ideas behind replicability might be valuable to study even in the non-linear deep RL setting. First, we evaluate our algorithm and its components on the well-studied CartPole environment (Barto et al., 1990). Then, we study the effects of quantized neural network Q-values in Atari environments (Bellemare et al., 2013).

### 5.1 EVALUATING REPLICABILITY ON REAL DATASETS

To show that our algorithms do not require impractically large amounts of data, we implement a version of fitted Q-iteration (Ernst et al., 2005) with replicable rounding akin to our generative model algorithm. We use the offline CartPole dataset available via d3rlpy (Seno & Imai, 2022) and a random Fourier feature encoding for $\phi$. Over 5 rounds, we use ridge regression to fit the value function. The rounding bin size is $\alpha = 0.2$.

We vary two components: To ensure that all policies are trained on distinct samples and to assess the amount of data needed for replicability, we sub-sample a fraction of the data for training. Then, we vary $\lambda$ to examine its impact. We evaluate the cumulative return as a measure of policy quality, and the largest fraction of identical learned weights across all runs. Figure 1 presents the results, averaged over 100 algorithm runs.

Our results show that replicability is achieved with a fraction of the available data and is correlated with high returns. This suggests that when the algorithm fails to fit the values, replication of policies becomes unlikely. While we expect regularization to play a role, its effect appears negligible here, likely because a few weights are disproportionately large. Available data seems to be the driver for replicability.

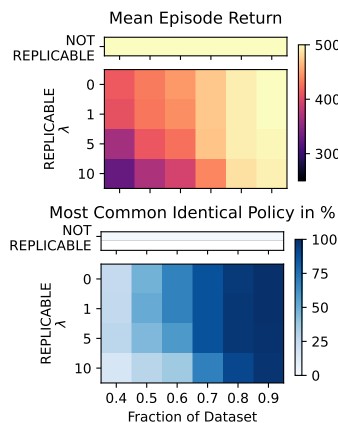

Figure 1: Mean return and percentage of most common identical weight vector. "Not replicable" indicates a baseline without regularization and rounding. Using a fraction of the data is sufficient to achieve replicability.

### 5.2 QUANTIZING NEURAL Q-VALUES

While we previously noted that achieving replicability in a neural network setting might be difficult without further work on feature learning, we set out to study the effects of our algorithmic elements on deep learning algorithms. We use the recent PQN algorithm (Gallici et al., 2025) to train Q-functions on

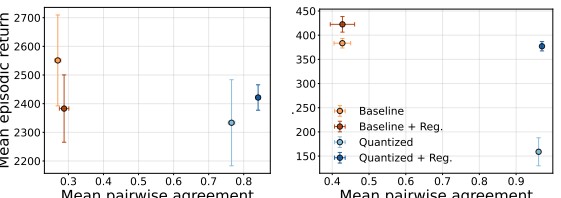

Figure 2: Mean return and agreement on MsPacman (left) Breakout (right). Quantization and regularization increase agreement while maintaining high performance.

MsPacman and Atari Breakout. Our theory suggests that rounding weights and using regularization can give rise to stability. The implication of rounded weights is rounded Q-values. Rather than rounding weights directly, which may lead to unforeseen challenges in deep learning, we round the outputs of our neural networks onto a fixed grid. We compare a version of quantized Q-values with regular PQN as well as regularized versions of both. We measure the final return by averaging each training run's 20 final returns. Then, we take holdout expert datasets from Minari (Younis et al., 2024) for the chosen tasks. On these, we compute the pairwise action agreement across seeds. We report mean and $1.96\times$ standard error over 15 runs in Figure 2 and hyperparameters in Appendix E.

Regularization alone is insufficient to ensure high agreement on either task while quantization leads to increased agreement. This is in part attributable to low action gaps (Farahmand, 2011) in the Atari games. While the quantized policies agree, they can do so on the wrong actions as indicated by the low return on Breakout. When combining regularization and quantization, the algorithm's return is within variance of the baseline, indicating no loss of performance. In addition, the benefits of agreement from quantization are kept providing empirical evidence for our theoretical findings.

## 6 RELATED WORK

**Linear Function Approximation RL**     Early asymptotic convergence guarantees for RL with function approximation were laid by Tsitsiklis & Van Roy (1996) while the study of *finite-sample* guarantees for RL with linear functions was initiated by Munos & Szepesvári (2008), who study the fitted value iteration algorithm with a generative model. Since then, various works have studied linearity in RL via a multitude of MDP assumptions (Jiang et al., 2017; Zanette et al., 2020b; Dann et al., 2018; Modi et al., 2020; Cai et al., 2020; He et al., 2021; Wang et al., 2021). Closely related to our work, others have studied version of linear MDPs that can represent mixture distributions (Jia et al., 2020; Ayoub et al., 2020; Zhou et al., 2021a; Zhou & Gu, 2022) or are represented via linear kernels (Zhou et al., 2021b). The linear MDP as studied in our paper was introduced by Yang & Wang (2019); Jin et al. (2020) and has since been studied quite extensively (Zanette et al., 2020a) where ultimately He et al. (2023) provide nearly minimax guarantees on the online problem. Reward free versions of both linear mixture MDPs (Chen et al., 2022; Zhang et al., 2023) as well as linear MDPs (Wagenmaker et al., 2022) have also been explored.

**Replicability**     The seminal idea of algorithmic stability has a long history in learning theory and given rise to various settings such as error stability (Kearns & Ron, 1999), uniform stability (Bousquet & Elisseeff, 2002), or differential privacy (Dwork et al., 2006), each quantifying how sensitive an algorithm's output is to changes in its input data. Recently notions of formal reproducibility have been proposed (Ahn et al., 2022; Impagliazzo et al., 2022). The notion we call replicability (Impagliazzo et al., 2022) was introduced to study the limits of stability. It asks that two executions of an algorithm on two different samples from the same distribution will yield the exact same outcome. Replicability is strongly related to aforementioned areas like privacy, or even generalization (Bun et al., 2023; Kalavasis et al., 2023). Since its inception, replicability has been studied for clustering (Esfandiari et al., 2023b), large-margin half spaces (Kalavasis et al., 2024a), hypothesis testing (Liu & Ye, 2024; Hopkins et al., 2024; Aamand et al., 2025), geometric partitions (Vander Woude et al., 2024), and online settings (Esfandiari et al., 2023a; Ahmadi et al., 2024; Komiyama et al., 2024). Hopkins & Moran (2025) study the role of randomness for replicability and Kalavasis et al. (2024b) provide an overview of the computational landscape of replicability. Closely related to ours is the work on replicable tabular RL (Eaton et al., 2023; Karbasi et al., 2023; Hopkins et al., 2025).

## 7 CONCLUSION AND FUTURE WORK

In this work, we provide algorithms for replicable ridge regression and uncentered covariance estimation as well as a set of algorithms for replicable RL with linear function approximation, both in the generative model and the episodic setting. Our experiments validate that the ideas introduced through replicability are feasible at real-world dataset sizes and that they extend naturally to the deep RL setting. Thus, our algorithms take a step towards building more reliable procedures that will facilitate safe deployment of RL in the wild. While we believe that this work can build the foundation for stable RL, there is no immediate societal impact as our manuscript is largely of theoretical nature.

We leave open several interesting questions for future work. Our algorithms were inspired by instability in deep learning but do not directly address the feature learning problem. Additionally, our experiments demonstrate how rounding via quantization can reduce policy differences in neural network training. Scaling these ideas to larger environments, including continuous spaces, is an important step toward ensuring replicability in real-world, safety-critical systems. Finally, concurrent work by Hopkins et al. (2025) shows that it is possible to achieve replicability in the tabular setting at little to no overhead cost. Given the sample complexity of the algorithms presented in this work, a core question is whether an approach like theirs can be extended to the linear setting as well.

ACKNOWLEDGMENTS

We gratefully acknowledge support from the Simons Foundation Collaboration on Algorithmic Fairness and the NSF ENCoRE TRIPODS Institute. EE and MH's research was partially supported by the DARPA Triage Challenge under award HR00112420305. Any opinions, findings, and conclusion or recommendations expressed in this material are those of the authors and do not necessarily reflect the view of DARPA or the US government.

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

# A PROOFS OF SECTION 3.1

## A.1 UNIFORM CONVERGENCE WITH INDEPENDENT, BUT NOT IDENTICALLY DISTRIBUTED DATA

Let $D_{[t]} = \{D_1, \ldots, D_t\}$ be a sequence of distributions and denote by $S \sim D_{[t]}^M$ a sample generated by taking $M$ i.i.d. draws from each of the $t$ distributions of $D_{[t]}$. Recall that $t$ denotes the round that algorithm is in and $M$ is the number of samples we draw per round. Note that $D_{[t]}^M$ is a distribution over a full sample of data of size $n = Mt$. Every data point drawn is independent from the other, but the data is only sampled from an identical distribution in blocks of size $M$. We prove below that independence is sufficient for proving Rademacher uniform convergence bounds with respect to $D_{[t]}^M$ by adapting the proof from Shalev-Shwartz & Ben-David (2014).

**Lemma A.1** (Expected Representativeness Bounded by Twice Rademacher Complexity; Independent Samples from Sequence of Distributions Version of Bartlett et al. (2005) Lemma A.5). *For a given sample $S \sim D_{[t]}^M$ of size $n = Mt$, let $L_S(\theta)) = \frac{1}{n} \sum_{i=1, z_i \in S}^n f(z_i)$ and let $L_{D_{[t]}^M}(\theta) = \mathbb{E}_S[L_S(\theta)]$.*

$$\mathbb{E}_{S \sim D_{[t]}^M} [\sup_\theta (L_{D_{[t]}^M}(\theta) - L_S(\theta))] \leq 2 \mathbb{E}_{S \sim D_{[t]}^M} [\frac{1}{n} \mathbb{E}_{\sigma \sim \{\pm 1\}^n} [\sup_\theta \sum_{i=1, z_i \in S}^n \sigma_i f(z_i)]]$$

*Proof.* Let $S' = \{z_1, \ldots, z_n'\} \sim D_{[t]}^M$ be another sample from the same distribution. For all $\theta \in \Theta$, $L_{D_{[t]}^M}(\theta) = \mathbb{E}_{S'}[L_{S'}(\theta)]$. Therefore for every $\theta$, we have that

$$L_{D_{[t]}^M}(\theta) - L_S(\theta) = \mathbb{E}_{S'}[L_{S'}(\theta) - L_S(\theta)].$$

If we take the supremum over both sides and then applying Jensen's inequality we get,

$$\sup_\theta (L_{D_{[t]}^M}(\theta) - L_S(\theta)) = \sup_\theta \mathbb{E}_{S'}[L_{S'}(\theta) - L_S(\theta)]$$
$$\leq \mathbb{E}_{S'}[\sup_\theta (L_{S'}(\theta) - L_S(\theta))].$$

Now, if we also take an expectation over the sample $S$, we get

$$\mathbb{E}_S[\sup_\theta (L_{D_{[t]}^M}(\theta) - L_S(\theta))] \leq \mathbb{E}_{S,S'}[\sup_\theta (L_{S'}(\theta) - L_S(\theta))]$$
$$= \frac{1}{n} \mathbb{E}_{S,S'}[\sup_\theta \sum_{i=1, z_i \in S, z_i' \in S'}^n (f(z_i') - f(z_i))]$$

Now, notice that for each $j$, $z_j$ and $z_j'$ are i.i.d. variables, with respect to each other. Notice that this does not rely on i.i.d. over all data points drawn over the sample, only over the data point drawn in each of $S$ and $S'$ coming from the same $D_j$ as i.i.d. samples. Therefore, within the expectation we can swap them out with each other (using the ghost sample trick) to get

$$\mathbb{E}_{S,S'}[\sup_\theta \left( ((f(z_j') - f(z_j)) + \sum_{i \neq j} (f(z_i') - f(z_i))) \right)] =$$
$$\mathbb{E}_{S,S'}[\sup_\theta \left( ((f(z_j) - f(z_j')) + \sum_{i \neq j} (f(z_i') - f(z_i))) \right)]$$

Now letting $\sigma_j$ be the random variable denoting $\mathbf{Pr}(\sigma_j = 1) = \mathbf{Pr}(\sigma_j = -1) = \frac{1}{2}$, we obtain that

$$\mathbb{E}_{S,S',\sigma_j}[\sup_\theta \left( (\sigma_j(f(z_j') - f(z_j)) + \sum_{i \neq j} (f(z_i') - f(z_i))) \right)] =$$
$$\mathbb{E}_{S,S'}[\sup_\theta \left( ((f(z_j) - f(z_j')) + \sum_{i \neq j} (f(z_i') - f(z_i))) \right)]$$

If we repeat this for all indices $j$, then we get that

$$\mathbb{E}_{S,S'}[\sup_\theta \sum_{i=1}^n (f(z_i') - f(z_i))] = \mathbb{E}_{S,S',\boldsymbol{\sigma}}[\sup_\theta \sum_{i=1}^n \sigma_i(f(z_i') - f(z_i))]$$

and using the fact that

$$\sup_\theta \sum_{i=1}^n \sigma_i(f(z_i') - f(z_i)) \leq \sup_\theta \sum_{i=1}^n \sigma_i f(z_i') + \sup_\theta \sum_{i=1}^n -\sigma_i f(z_i')$$

Finally, we can upper bound

$$\mathbb{E}_{S,S',\boldsymbol{\sigma}}[\sup_\theta \sum_{i=1}^n \sigma_i(f(z_i') - f(z_i)) \leq \sup_\theta \sum_{i=1}^n \sigma_i f(z_i') + \sup_\theta \sum_{i=1}^n \sigma_i f(z_i')]$$

$$2 \mathbb{E}_{S \sim D_{[t]}^M}[\frac{1}{n} \mathbb{E}_{\sigma \sim \{\pm 1\}^n}[\sup_\theta \sum_{i=1,z_i \in S}^n \sigma_i f(z_i)]]$$

$\square$

## A.2 Gradient concentration and strong convexity

We want to prove the replicability of Algorithm 2. To do so, we will show that the empirical minimizer induced by the algorithm is close to the expected minimizer in L2 norm via convexity. This we will use to obtain a bound on the difference between estimator produced by two independent runs of the algorithm. Define

$$R(\theta) = \mathbb{E}_{S \sim D_{[t]}^m}[\sum_{(\mathbf{x},y) \in S} (\langle \theta, \mathbf{x} \rangle - y)^2] + \lambda \|\theta\|_2^2$$

and let

$$\widehat{R}_S(\theta) = \sum_{(\mathbf{x},y) \in S} (\langle \theta, \mathbf{x} \rangle - y)^2 + \lambda \|\theta\|_2^2.$$

First, we will prove uniform convergence of the gradient difference of these two functions. To get to this result we will build on a result by Foster et al. (2018) who provide bounds for the uniform convergence of gradients in non-convex learning. While this tools are more general than what we need, it still gives us dimension-free bounds on the ridge regression gradient. The key statement that we will need is Proposition A.1. The original statement by Foster et al. (2018) provides guarantees for i.i.d. data. The i.i.d. ness of the data goes back to Lemma A.5 by (Bartlett et al., 2005). We reprove this Lemma with our data requirements in Lemma A.1. The remaining elements that are used in the proof of proposition A.1 are Theorem A.2 in (Bartlett et al., 2005) and Lemma 4 in (Foster et al., 2018) which still hold. We thus state the slightly generalized form of Proposition 2 by (Foster et al., 2018) here:

**Proposition A.1** (Symmetrization ((Foster et al., 2018))). *Let $L_D(\theta) = \mathbb{E}_{(x,y) \sim D_{[t]}^M}[\ell(\theta; x, y)]$ denote some expected risk function parametrized by some weight vector $\theta$. Let $\hat{L}$ be the corresponding empirical risk function. For any $\delta > 0$, with probability at least $1 - \delta$ over the independent draw of data $\{x_i, y_i\}_{i=0}^{N-1}$,*

$$\mathbb{E}\sup_\theta \|\nabla L_D(\theta) - \nabla \hat{L}(\theta)\| \leq \frac{4}{N} E_\sigma \sup_\theta \left\|\sum_{i=1}^{N-1} \sigma_i \nabla \ell(\theta; x_i, y_i)\right\| + \sup_{\theta,x,y} \|\nabla \ell(\theta; x, y)\| \frac{\log 1/\delta}{N}$$

To bound the first term on the RHS, the following Theorem is then introduced

**Theorem A.1** (Rademacher Chain Rule (Foster et al., 2018)). *Let sequences of functions $G_i : \mathbb{R}^K \mapsto \mathbb{R}$ and $F_i : \mathbb{R}^d \mapsto \mathbb{R}^K$ be given. Suppose there are constants $L_G$ and $L_F$ s.t. for all $1 \leq i \leq N$, $\|\nabla G_i\| \leq L_G$ and $\sqrt{\sum_{k=0}^{K-1} \|\nabla F_{i,j}(w)\|^2} \leq L_F$. Then*

$$\frac{1}{2} \mathbb{E}_{\sigma_i} \sup_\theta \left\|\sum_{i=0}^{N-1} \sigma_i \nabla(G_i(F_i(\theta)))\right\| \leq L_F \mathbb{E}_{\sigma_i} \sup_\theta \sum_{i=0}^{N-1} \langle \sigma_i, \nabla G_i(F_i(\theta)) \rangle + L_G \mathbb{E}_{\sigma_i} \sup_\theta \left\|\sum_{i=0}^{N-1} F_i(\theta)\sigma_i\right\|$$

*where $\nabla F_i$ is the Jacobian of $F_i$ which lives in $R^{d \times K}$ and $\sigma \in \{\pm 1\}^{K \times N}$ is a matrix of Rademacher random variables.*

An immediate consequence of this lemma is the uniform convergence guarantee of the ridge regression gradient which we prove here.

**Theorem A.2** (Uniform Convergence of the Ridge Gradient). *Let $\{(x_i, y_i)\}_{i=0}^{N-1} \subseteq \mathbb{R}^d \times \mathbb{R}$ be i.i.d. samples drawn from a distribution D, with $\|x_i\|_2 \leq 1$ and $|y_i| \leq Y$. For a fixed radius $B \geq 0$, define the function class*

$$\mathcal{F} = \left\{ (x, y) \mapsto (\theta^\top x - y)^2 \ : \ \|\theta\|_2 \leq B \right\}.$$

*Then there exists an absolute constant $c > 0$ such that if*

$$N \in \Omega \left( \frac{(B + Y)^2}{\varepsilon^2} \log \frac{1}{\delta} \right)$$

*then with probability at least $1 - \delta$,*

$$\sup_\theta \left\| \nabla_\theta R(\theta) - \nabla_\theta \widehat{R}_S(\theta) \right\| \leq \varepsilon.$$

*Proof.* The outline of the proof is as follows. First, we obtain a and upper bound on the expected supremum norm of the gradient via the vector valued Rademacher statements in Proposition A.1 and Theorem A.1, then we conclude a total bound on the number of samples required via McDiarmid. However, before we do so, we note the following. In the gradient formulation of the ridge regressor, the weight regularization term is independent from the data and simply cancels out in the difference of the gradients of $\nabla R(\theta) - \nabla \widehat{R}_S(\theta)$. As a consequence, it suffices to bound only the data dependent term. Thus, we start the proof by instantiating the loss as $\ell = \frac{1}{2}(\theta^\top x - y)^2 + \lambda \|\theta\|^2$ which means by Proposition A.1

$$\mathbb{E} \sup_\theta \|\nabla R(\theta) - \nabla \widehat{R}_S(\theta)\| \leq \frac{4}{N} \mathbb{E}_\sigma \sup_\theta \left\| \sum_{i=1}^{N-1} \sigma_i \nabla \ell(\theta; x_i, y_i) \right\| + \sup_{\theta, x, y} \|\nabla \ell(\theta; x, y)\| \frac{\log 1/\delta_1}{N}$$

The gradient of the loss can be written as $\nabla \ell(\theta; x, y) = (\theta^\top x - y)x$. Since the norms of all elements in the supremum are bounded the term on the right is also easily bounded

$$\sup_{\theta, x, y} \|\nabla \ell(\theta; x, y)\| = \sup_{\theta, x, y} \left\| (\theta^\top x - y)x \right\| \leq (B + Y)$$

It remains to bound the first term on the RHS. We invoke Theorem A.1 with $G(a) = \frac{1}{2}(a - y)^2$, $F(\theta) = (\theta^\top x)$ and $k = 1$. Suppose $\|a\| \leq A$ then $G(a)$ is A-Lipshitz. More precisely, we will have that $\sup_{a, y} |G'(a)| \leq (B + Y)$. Furthermore $\nabla F(\theta) = x$ and we know that $\|x\| \leq 1$ which means that $L_F = 1$. As a result, we have

$$\mathbb{E}_\sigma \sup_\theta \left\| \sum_{i=1}^{N-1} \sigma_i \nabla \ell(\theta; x_i, y_i) \right\| \leq \mathbb{E}_\sigma \left[ \sup_\theta \sum_{i=1}^{N-1} \sigma_i G_i'(\theta^\top x_i) \right] + A \mathbb{E}_\sigma \left\| \sum_{i=0}^{N-1} \sigma_i x_i \right\|$$

$$\leq \mathbb{E}_\sigma \left[ \sup_\theta \sum_{i=1}^{N-1} \sigma_i (\theta^\top x_i - y_i) \right] + A \mathbb{E}_\sigma \left\| \sum_{i=0}^{N-1} \sigma_i x_i \right\|$$

$$= \mathbb{E}_\sigma \left[ \sup_\theta \sum_{i=1}^{N-1} \sigma_i \theta^\top x_i \right] - \mathbb{E}_\sigma \left[ \sum_{i=1}^{N-1} \sigma_i y_i \right] + A$$

$$E_\sigma \left\| \sum_{i=0}^{N-1} \sigma_i x_i \right\|$$

$$\leq B \mathbb{E}_\sigma \left\| \sum_{i=1}^{N-1} \sigma_i x_i \right\| + (B + Y) \mathbb{E}_\sigma \left\| \sum_{i=0}^{N-1} \sigma_i x_i \right\|$$

$$\leq 2(B + Y)\sqrt{N}$$

where the last step is a standard Rademacher argument (see, e.g. (Shalev-Shwartz & Ben-David, 2014)) It remains to move from the expected value bound to a high probability bound. So far, we have

$$\mathbb{E} \sup_\theta \|\nabla R(\theta) - \nabla \widehat{R}_S(\theta)\| \leq \frac{4 \times 2(B + Y)\sqrt{N}}{N} + (B + Y) \frac{\log(1/\delta_1)}{N}$$

$$\leq \frac{4 \times 2(B+Y)\sqrt{N}}{N} + (B+Y)\sqrt{\frac{\log(1/\delta_1)}{N}}$$

$$\leq \frac{(B+Y)(8 + \sqrt{\log(1/\delta_1)})}{\sqrt{N}}$$

where the second inequality holds as long as we pick $N > \log(1/\delta)$, s.t. $\frac{\log(1/\delta)}{N} \leq 1$. Observe that the bounded difference $\sup_\theta \|\nabla R(\theta) - \nabla \widehat{R}_S(\theta)\|$ changes by at most $(B+Y)/N$ if we swapped out one sample in the empirical average since $\|\ell(\theta; x, y) \leq B+Y\|$. As a result, we have by McDiarmid's inequality that

$$\mathbf{Pr}\left[\sup_\theta \left\|\nabla R(\theta) - \nabla \widehat{R}_S(\theta)\right\| > \mathbb{E}\left[\sup_\theta \left\|\nabla R(\theta) - \nabla \widehat{R}_S(\theta)\right\|\right] + t\right] \leq \exp\left(-\frac{2Nt^2}{(B+Y)^2}\right) \leq \delta_2$$

By setting $\delta_1 = \delta_2 = \delta/2$ and applying a union bound, we have with probability $1 - \delta$ that

$$\sup_\theta \left\|\nabla R(\theta) - \nabla \widehat{R}_S(\theta)\right\| \leq \frac{(B+Y)(8 + \sqrt{\log(2/\delta)} + \sqrt{\log(1/\delta)})}{\sqrt{N}} \leq \frac{(B+Y)(8 + 2\sqrt{\log(2/\delta)})}{\sqrt{N}}$$

Setting equal to $\varepsilon$ and solving yields

$$\frac{(B+Y)^2(8 + 2\sqrt{\log(2/\delta)})^2}{\varepsilon^2} \leq \frac{100(B+Y)^2}{\varepsilon^2}\log\frac{2}{\delta} \leq N$$

$\square$

**Lemma A.2** (Parameter Bound for Ridge via Strong Convexity). *Suppose $\lambda > 0$ and $\|\theta\|_2 \leq B$. We define*

$$\widetilde{\theta} = \arg\min_\theta R(\theta), \qquad \hat{\theta} = \arg\min_\theta \widehat{R}_S(\theta).$$

*Because $R(\theta)$ is $2\lambda$-strongly convex, it has a unique minimizer $\widetilde{\theta}$. Conditioned on the fact that*

$$\sup_\theta \left\|\nabla_\theta R(\theta) - \nabla_\theta \widehat{R}_S(\theta)\right\| \leq \varepsilon.$$

*we can bound the parameters of the estimator as*

$$\|\hat{\theta} - \widetilde{\theta}\|_2 \leq \frac{\varepsilon}{2\lambda}.$$

*Proof.* Note that both $R$ and $\widehat{R}_S$ are $2\lambda$ strongly convex. Consequently, the unique minimizers satisfy $\nabla R(\widetilde{\theta}) = 0 = \nabla \widehat{R}_S(\hat{\theta})$. and we have that $\|\nabla R(\hat{\theta}) - \nabla \widehat{R}_S(\hat{\theta})\| = \|\nabla R(\hat{\theta})\| \leq \varepsilon$. Now, by strong convexity we have

$$2\lambda\|\widetilde{\theta} - \hat{\theta}\|^2 \leq (\nabla R(\widetilde{\theta}) - \nabla R(\hat{\theta}))^T(\widetilde{\theta} - \hat{\theta}) \leq \|\nabla R(\widetilde{\theta}) - \nabla R(\hat{\theta})\|\|(\widetilde{\theta} - \hat{\theta})\|$$

When $\widetilde{\theta} = \hat{\theta}$ the inequality holds. Thus, we can safely divide both sides by the norm of the parameter vectors and we get.

$$2\lambda\|\widetilde{\theta} - \hat{\theta}\| \leq \|\nabla R(\widetilde{\theta}) - \nabla R(\hat{\theta})\|$$

Recall that $\nabla R(\widetilde{\theta}) = 0$, so we have

$$2\lambda\|\widetilde{\theta} - \hat{\theta}\| \leq \|\nabla R(\hat{\theta})\| \leq \varepsilon$$

$$\implies \|\widetilde{\theta} - \hat{\theta}\| \leq \frac{\varepsilon}{2\lambda}$$

$\square$

## A.3 PROOF OF THEOREM 3.1

With these tools equipped we are ready to prove Theorem 3.1.

*Proof.* Conditioned on the success of Theorem A.2 and Lemma A.2, we know that

$$\|\hat{\theta} - \widetilde{\theta}\| \leq \varepsilon'/(2\lambda) = \frac{\Delta}{2}.$$

Consider now two iterations of the same procedure producing two estimates $\hat{\theta}^{(1)}$ and $\hat{\theta}^{(2)}$. By triangle inequality and the above, we have that these two estimates can differ by at most $2\|\hat{\theta} - \widetilde{\theta}\|$ which means that

$$\|\hat{\theta}^{(1)} - \hat{\theta}^{(2)}\| \leq \varepsilon'/\lambda = \Delta.$$

Now, by Lemma 2.1, our rounding procedures maps these two vectors onto the same vector on the grid with probability $1 - d\frac{\Delta}{\alpha}$. For the error of each *rounded* estimate, we have

$$\|\bar{\theta} - \widetilde{\theta}\| \leq \|\bar{\theta} - \hat{\theta}\| + \|\hat{\theta} - \widetilde{\theta}\| = \sqrt{d}\frac{\alpha}{2} + \frac{\Delta}{2}$$

By choosing $\alpha = \dfrac{d\varepsilon}{d^{3/2} + \rho - 2\delta}$ we account for the $2\delta$ probability of failure across two independent algorithm executions and by choosing $\varepsilon'$ such that $\Delta \leq \frac{\varepsilon(\rho - 2\delta)}{d^{3/2} + \rho - 2\delta}$, we have that

$$d\frac{\Delta}{\alpha} = d\frac{d^{3/2} + \rho - 2\delta}{d\varepsilon} \times \frac{\varepsilon(\rho - 2\delta)}{d^{3/2} + \rho - 2\delta} = \rho - 2\delta.$$

Furthermore, we satisfy

$$\sqrt{d}\frac{\alpha}{2} + \frac{\Delta}{2} = \frac{\sqrt{d}d\varepsilon}{2(d^{3/2} + \rho - 2\delta)} + \frac{\varepsilon(\rho - 2\delta)}{2(d^{3/2} + \rho - 2\delta)} = \frac{\varepsilon\left(d^{3/2} + \rho - 2\delta\right)}{2\left(d^{3/2} + \rho - 2\delta\right)} \leq \varepsilon$$

Finally, we need to find $\varepsilon'$ to obtain our sample complexity. We have that

$$\frac{\varepsilon'}{2\lambda} = \frac{\varepsilon(\rho - 2\delta)}{d^{3/2} + \rho - 2\delta}$$

$$\iff \quad \varepsilon' = \frac{2\lambda\varepsilon(\rho - 2\delta)}{d^{3/2} + \rho - 2\delta}$$

Plugging $\varepsilon'$ into

$$N \geq \frac{100(B + Y)^2}{\varepsilon'^2} \log\left(\frac{2}{\delta}\right)$$

yields

$$\frac{100(B + Y)^2(d^{3/2} + \rho - 2\delta)^2}{4\lambda^2\varepsilon^2(\rho - 2\delta)^2} \log\frac{2}{\delta} \leq \frac{25(B + Y)^2 d^3}{\lambda^2\varepsilon^2(\rho - 2\delta)^2} \log\frac{2}{\delta} \quad \leq N$$

□

## A.4 PROOF OF THEOREM 3.2

*Proof.* Note that our assumptions do not change anything about the replicability of the estimator. As such it remains to prove the accuracy guarantee. We want to prove that under the stated assumptions it holds that

$$\max_{\mathbf{x}} |\mathbf{x}^T(\overline{\mathbf{w}} - \theta^*)| \leq \varepsilon.$$

Note that we can easily decompose the inner term via triangle inequality

$$|\mathbf{x}^T(\hat{\theta} - \theta^*)| \leq |\mathbf{x}^T(\overline{\mathbf{w}} - \widetilde{\theta})| + |\mathbf{x}^T(\widetilde{\theta} - \theta^*)|$$

By data assumption and our Theorem 3.2, we can have that $|\mathbf{x}^T(\overline{\mathbf{w}} - \widetilde{\theta})| \leq \|x\|\|(\overline{\mathbf{w}} - \widetilde{\theta})\| \leq \dfrac{\varepsilon}{2}$. It remains to prove that $|\mathbf{x}^T(\widetilde{\theta} - \theta^*)|$ is small. We can use the core-set assumption to rewrite this term as follows

$$
\begin{aligned}
\left|\mathbf{x}^T(\widetilde{\theta} - \theta^*)\right| &= \left|\sum_i \eta_i \nu(\mathbf{x}_i)\mathbf{x}_i^\top(\widetilde{\theta} - \theta^*)\right| \\
&\leq \|\eta\|\left\|\sum_i \nu(\mathbf{x}_i)\mathbf{x}_i^\top(\widetilde{\theta} - \theta^*)\right\| \\
&\leq \sqrt{k}\sqrt{\sum_i \nu(\mathbf{x}_i)^2(\mathbf{x}_i^\top(\widetilde{\theta} - \theta^*))^2} \\
&\leq \sqrt{k}\sqrt{\mathop{\mathbb{E}}_{x\sim\nu}\left[(\mathbf{x}_i^\top(\widetilde{\theta} - \theta^*))^2\right]}
\end{aligned}
$$

At this point, it remains to show that $\mathbb{E}_{x\sim\nu}\left[(\mathbf{x}_i^\top(\widetilde{\theta} - \theta^*))^2\right]$ is small. By definition, we know that $\widetilde{\theta} = \arg\min \mathbb{E}_{\mathbf{x},y}[(\theta^\top\mathbf{x} - y)^2 + \lambda\|\theta\|^2]$. Since $y = \theta^{*\top}\mathbf{x} + \epsilon$, we have that

$$
\begin{aligned}
\mathop{\mathbb{E}}_{\mathbf{x}\sim\nu}\left[(\mathbf{x}_i^\top(\widetilde{\theta} - \theta^*))^2\right] + \lambda\|\theta\|^2 &\leq \mathop{\mathbb{E}}_{\mathbf{x},y}\left[(\mathbf{x}^\top\theta^* - y)^2\right] + \lambda\|\theta^*\|^2 \\
&= \mathop{\mathbb{E}}_{\mathbf{x},\epsilon}\left[(\mathbf{x}^\top\theta^* - \theta^{*\top}\mathbf{x} - \epsilon)^2\right] + \lambda\|\theta^*\|^2 = \lambda\|\theta^*\|^2
\end{aligned}
$$

Plugging this back in we get

$$
\left|\mathbf{x}^T(\widetilde{\theta} - \theta^*)\right| \leq \sqrt{k\lambda}\|\theta^*\|
$$

Finally, choosing $\lambda = \dfrac{\varepsilon^2}{4k\|\theta^*\|^2}$ yields that $\left|\mathbf{x}^T(\widetilde{\theta} - \theta^*)\right| \leq \varepsilon/2$ $\qquad\square$

# B PROOFS OF SECTION 3.2

## B.1 PROOF OF THEOREM 3.3

*Proof.* To prove the Theorem, we begin by obtaining an elementwise bound against the expected covariance matrix. Let

$$\widehat{\Sigma}_{jl} := \sum_{i=0}^{t-1} \frac{1}{M} \sum_{m=0}^{M-1} \mathbf{x}_{i,m}^j \mathbf{x}_{i,m}^l, \qquad \Sigma_{jl} := \sum_{t=0}^{t-1} \mathop{\mathbb{E}}_{D_t} \left[ \mathbf{x}^j \mathbf{x}^l \right].$$

By Hoeffding, a union bound, and $\|\mathbf{x}\| \le 1$,

$$\mathbf{Pr} \left[ \bigcup_{1 \le j \le l \le d} \left| \widehat{\Sigma}_{jl} - \Sigma_{jl} \right| \ge \tau \right] \le 2d^2 \exp \left( -\frac{M\tau^2}{2t} \right) \le \delta.$$

Thus, setting $\tau = \varepsilon'/d$, when we draw at least

$$\frac{2td^2}{\varepsilon'^2} \log \frac{2d^2}{\delta} \le M$$

samples per distribution, we obtain with high probability that

$$\|\widehat{\Sigma} - \Sigma\|_F \le \varepsilon'.$$

Conditioned on success of estimation, two such estimates can differ by at most by $\|\widehat{\Sigma}^{(1)} - \widehat{\Sigma}^{(2)}\|_F \le 2\varepsilon' = \Delta$.

We are interested in the replicability and accuracy after rounding. By Lemma 2.1, we want that $d^2 \Delta / \alpha \le \rho - 2\delta$. Furthermore we want for both estimates that

$$\|\Pi_{\mathrm{PSD}}(\overline{\Sigma}) - \Sigma\|_F \le \varepsilon$$

Note that the eigenvalue clipping is simply the metric projection onto the PSD cone $\mathbb{S}_+^d$ in the Hilbert space $(\mathbb{S}^d, \langle \cdot, \cdot \rangle_F)$ that the original covariance lies in. Metric projections onto closed convex sets in Hilbert spaces are nonexpansive (i.e., 1-Lipschitz; see, e.g., (Bauschke & Combettes, 2017)). In addition, since $\Sigma \in \mathbb{S}_+^d$, our clipping operator would not change the true covariance matrix at all if applied and we have $\Pi_{\mathrm{PSD}}(\Sigma) = \Sigma$. As a result, we have that

$$\|\Pi_{\mathrm{PSD}}(\overline{\Sigma}) - \Sigma\|_F = \|\Pi_{\mathrm{PSD}}(\overline{\Sigma}) - \Pi_{\mathrm{PSD}}(\Sigma)\|_F \le \|\overline{\Sigma} - \Sigma\|_F$$

Thus, it is sufficient to show that the rounded matrix before projection does not incur large error. We do this by decomposing as follows.

$$\|\overline{\Sigma} - \Sigma\|_F \le \|\overline{\Sigma} - \widehat{\Sigma}\|_F + \|\widehat{\Sigma} - \Sigma\|_F \le \varepsilon.$$

By setting $\alpha = \dfrac{d^2 \varepsilon}{(d^3 + \rho - 2\delta)}$ we account for the $2\delta$ probability of failure across two independent algorithm executions and by setting $\varepsilon'$ such that $\Delta = \dfrac{\varepsilon(\rho - 2\delta)}{(d^3 + \rho - 2\delta)}$ where we account for the failure probability of two executions, we satisfy

$$d^2 \frac{\Delta}{\alpha} = d^2 \frac{\varepsilon(\rho - 2\delta)}{(d^3 + \rho - 2\delta)} \frac{(d^3 + \rho - 2\delta)}{d^2 \varepsilon} \le \rho - 2\delta$$

as well as

$$d\frac{\alpha}{2} + \frac{\Delta}{2} = d\frac{d^2 \varepsilon}{2(d^3 + \rho - 2\delta)} + \frac{\varepsilon(\rho - 2\delta)}{2(d^3 + \rho - 2\delta)} = \frac{\varepsilon(d^3 + \rho - 2\delta)}{2(d^3 + \rho - 2\delta)} \le \varepsilon.$$

The total sample complexity after plugging in $\varepsilon'$ comes to

$$\frac{4td^2(d^3 + \rho - 2\delta)^2}{2\varepsilon^2(\rho - 2\delta)^2} \log \frac{2d^2}{\delta} \le \frac{8td^8}{\varepsilon^2(\rho - 2\delta)^2} \log \frac{2d^2}{\delta} \le M$$

Accounting for $t$ distributions finishes the proof. $\qquad\square$

## C   PROOFS FOR SECTION 4.1

We provide additional proofs required to complete the proof in the main section here. The following is a standard result from the literature that we restate for completeness (see e.g. (Kakade, 2003) for a similar argument).

**Lemma C.1.** *Let $\mathcal{T}_h Q_{h+1} := R_h(s,a) + P_h \max_a Q_{h+1}(s,a)$ denote the standard Bellman operator. Assume that for all h, we have*

$$\|\hat{Q}_h - \mathcal{T}_h \hat{Q}_{h+1}\|_\infty \leq \varepsilon.$$

*Then we have*

- *Accuracy of $\hat{Q}_h$: $\forall h$, $\|\hat{Q}_h - Q_h^*\| \leq (H-h)\varepsilon$*
- *Policy performance: for $\hat{\pi}_h(s) := \arg\max_a \hat{Q}_h(s,a)$, then we have $|V^{\hat{\pi}} - V^*| \leq 2H^2\varepsilon$.*

*Proof.* **First Claim:** By backward induction on h.
Base case: Starting from $Q_H(s,a) = 0$, we have $\mathcal{T}_{H-1}Q_H(s,a) = r$. By our assumption, this implies that $\hat{Q}_{H-1} - r \leq \varepsilon$. As a result, we know that $\|\hat{Q}_{H-1} - Q_{H-1}^*\|_\infty \leq \varepsilon$
Inductive step: Our inductive hypothesis now states $\|\hat{Q}_{h+1} - Q_{h+1}^*\| \leq (H-h-1)\varepsilon$. We have

$$\begin{aligned}
|\hat{Q}_h(s,a) - Q^*(s,a)| &\leq |\hat{Q}_h - \mathcal{T}_h \hat{Q}_{h+1}(s,a)| + |\mathcal{T}_h \hat{Q}_{h+1}(s,a) - Q_h^*(s,a)| \\
&\leq \varepsilon + |\mathcal{T}_h \hat{Q}_{h+1}(s,a) - Q_h^*(s,a)| \\
&\leq \varepsilon + (H-h-1)\varepsilon \leq (H-h)\varepsilon
\end{aligned}$$

**Second Claim:** Again by backward induction starting at $H-1$.
Base case: For any s,

$$\begin{aligned}
V_{H-1}^{\hat{\pi}}(s) - V_{H-1}^*(s) &= \hat{Q}_{H-1}(s, \hat{\pi}_{H-1}(s)) - Q_{H-1}^*(s, \pi_{H-1}^*(s)) \\
&= \hat{Q}_{H-1}(s, \hat{\pi}_{H-1}(s)) - Q_{H-1}^*(s, \hat{\pi}_{H-1}(s)) \\
&\quad + Q_{H-1}^*(s, \hat{\pi}_{H-1}(s)) - Q_{H-1}^*(s, \pi_{H-1}^*(s)) \\
&= Q_{H-1}^*(s, \hat{\pi}_{H-1}(s)) - Q_{H-1}^*(s, \pi_{H-1}^*(s)) \\
&\geq Q_{H-1}^*(s, \hat{\pi}_{H-1}(s)) - \hat{Q}_{H-1}(s, \hat{\pi}_{h-1}(s)) \\
&\quad + \hat{Q}_{H-1}(s, \pi_{H-1}^*(s)) - Q_{H-1}^*(s, \pi_{H-1}^*(s)) \\
&\geq 2\varepsilon
\end{aligned}$$

The third equality here uses the fact that $\hat{Q}_{H-1}(s,a) = Q_{H-1}^*(s,a) = R(s,a)$ and the first inequality uses the fact that within our estimated Q-function, we always pick the action with the largest Q-value when following our policy. The last step then uses the first claim.

Inductive step: Our induction hypothesis states that $V_{h+1}^{\hat{\pi}}(s) - V_{h+1}^*(s) \geq -2(H-h-1)H\varepsilon$. We have that

$$\begin{aligned}
V_h^{\hat{\pi}}(s) - V_h^*(s) &= \hat{Q}_h(s, \hat{\pi}_h(s)) - Q_h^*(s, \pi_h^*(s)) \\
&= \hat{Q}_h(s, \hat{\pi}_h(s)) - Q_h^*(s, \hat{\pi}_h(s)) + Q_h^*(s, \hat{\pi}_h(s)) - Q_h^*(s, \pi_h^*(s)) \\
&= \mathop{\mathbb{E}}_{s \sim P_h(s, \hat{\pi}_h(s))}[V_{h+1}^{\hat{\pi}}(s) - V_{h+1}^*(s)] + Q_h^*(s, \hat{\pi}_h(s)) - Q_h^*(s, \pi_h^*(s)) \\
&\geq -2(H-h-1)H\varepsilon + Q_h^*(s, \hat{\pi}_h(s)) - \hat{Q}_h(s, \hat{\pi}_h(s)) \\
&\quad + \hat{Q}_h(s, \pi_h^*(s)) - Q_h^*(s, \pi_h^*(s)) \\
&\geq -2(H-h-1)H\varepsilon - 2(H-h)\varepsilon \geq -2(H-h)H\varepsilon
\end{aligned}$$

$\square$

### C.1   PROOF OF THEOREM 4.1

*Proof.* The proof builds on the result in Theorem 3.2. We make the following observations. In every iteration, we call a ridge regression procedure from $\phi(s_h, a_h)$ to the target variable $R_h(s_h, a_h) +$

$\max_a \hat{Q}_{h+1}(s_{h+1}, a)$. At every step, we know that the Bayes optimal predictor is defined through the Bellman operator $\mathcal{T}$ as

$$\mathbb{E}_{s_{h+1}\sim P_h} [R_h(s_h, a_h) + \max_{a'} \hat{Q}_{h+1}(s_{h+1}, a')] = \mathcal{T}_h(\hat{\theta}_{h+1})^\top \phi(s_h, a_h).$$

Also, note that for all $(s, a)$ the expected value of the per state-action pair noise $\epsilon_{(s,a)}$ on the predictor is $\mathbb{E}_{s'\sim P}[\epsilon_{(s,a)}] = \mathbb{E}_{s'\sim P}[R(s, a) + \max_{a'} \hat{Q}_{h+1}(s', a) - \mathcal{T}_h(\hat{Q}_h(s, a))] = 0$. We immediately have from Theorem 3.2 that for all $h$

$$\max_{s_h, a_h} |\phi(s_h, a_h)^T (\hat{\theta}_h - \mathcal{T}_h(\hat{\theta}_{h+1}))| \le \varepsilon'$$

with failure probability $\delta'$ and replicability parameter $\rho'$. We now need to union bound over the number of rounds and make sure our error does not exceed $\varepsilon$ in total. Since we are running exactly $H$ rounds, it suffices to choose $\delta' = \delta/H$ and $\rho' = \rho/H$. For accuracy, we choose $\varepsilon' = \varepsilon/(2H^2)$. As a result, we have for all $h \in H$ that $\|\hat{Q}_h(s_h, a_h) - \mathcal{T}_h\hat{Q}_{h+1}(s_h, a_h)\|_\infty \le \varepsilon/(2H^2)$ with probability $1 - \delta$. By Lemma C.1, we immediately have that $|V^*(s) - V^\pi(s)| \le \varepsilon$. Let us denote $c_1$ the constant term in the cost of the ridge regression procedure. At this point, we note that the ground truth parameters of the optimal policy are bounded as $\|\mathbf{w}_h^\pi\| \le 2H\sqrt{d}$ via Proposition 2.1. It remains to show that norm of the weights output by our algorithm are within a bounded ball B. Recall that we are solving a ridge regression problem of the form

$$\frac{1}{M} \sum_{m\in M} (\hat{\mathbf{w}}_h^\top \phi(s_{m,h}, a_{m,h}) - (R_h(s_{m,h}, a_{m,h}) + V_{h+1}(s_{m,h+1}))^2 + \lambda\|\hat{\mathbf{w}}_h\|^2$$

Using optimality of $\hat{\mathbf{w}}_h$ we can compare to the zero vector. Since $\hat{\mathbf{w}}_h$ minimizes our objective we have

$$\frac{1}{M} \sum_{m\in M} (\hat{\mathbf{w}}_h^\top \phi(s_{m,h}, a_{m,h}) - (R_h(s_{m,h}, a_{m,h}) + V_{h+1}(s_{m,h+1}))^2 + \lambda\|\hat{\mathbf{w}}_h\|^2$$

$$\le \frac{1}{M} \sum_{m\in M} (\vec{0}^\top \phi(s_{m,h}, a_{m,h}) - (R_h(s_{m,h}, a_{m,h}) + V_{h+1}(s_{m,h+1}))^2 + 0 \le 4H^2$$

which implies

$$\lambda\|\hat{\mathbf{w}}_h\|^2 \le 4H^2 \Rightarrow \|\hat{\mathbf{w}}_h\| \le \frac{2H}{\sqrt{\lambda}}$$

From the proof of Theorem 3.2, we know that $\lambda = \dfrac{\varepsilon^2}{4k\|\theta^*\|^2}$. That means, it will suffice to choose

$$\|\hat{\mathbf{w}}_h\| \le B = \frac{4\sqrt{k}H\|\theta^*\|}{\varepsilon} = \frac{8\sqrt{k}\sqrt{d}H^2}{\varepsilon} \tag{1}$$

Our total sample complexity is

$$H \sum_{(s,a)} \lceil \nu(s, a)M \rceil \le H \sum_{(s,a)\in C_k} (1 + \nu(s, a)M)$$

$$\le H \left( k + \frac{c_1 16 \left(\frac{8\sqrt{k}\sqrt{d}H^2}{\varepsilon} + 2H\right)^2 d^5 k^2 H^{18}}{\varepsilon^6(\rho - 2\delta)^2} \log\left(\frac{H}{\delta}\right) \right)$$

$$\le H \left( k + \frac{c_1 16 \left(\frac{10\sqrt{k}\sqrt{d}H^2}{\varepsilon}\right)^2 d^5 k^2 H^{18}}{\varepsilon^6(\rho - 2\delta)^2} \log\left(\frac{H}{\delta}\right) \right)$$

$$\leq H \left( k + \frac{c_1 1600 d^6 k^3 H^{22}}{\varepsilon^8 (\rho - 2\delta)^2} \log \left( \frac{H}{\delta} \right) \right)$$

This finishes the accuracy part of the proof.

It remains to prove replicability. We prove replicability of the procedure by backward induction. Note that all values are initialized to $0$ which is always replicable, so the base case holds. Suppose now that the estimate in round $h+1$ of $V_{h+1}$ was replicable. Since the rewards are deterministic, and $V_{h+1}$ was replicable, the label distribution of our ridge regressor is the same in round $V_h$. The estimate of $\mathbf{w}_H^\top$ is thus replicable with probability $\rho/H$. Since there are only $H$ rounds, the total procedure is replicable with probability $\rho$. □

# D  PROOF OF SECTION 4.2

## D.1  PROOFS FOR THEOREM 4.2

In the following, let $\Lambda_h^t = \sum_{i \in [t]} \sum_{m \in [M]} \phi(s_{m,h}^i, a_{m,h}^i) \phi(s_{m,h}^i, a_{m,h}^i)^T + \lambda I$ be the regularized Gram matrix used for ridge regression. Denote the ridge solution by

$$\mathbf{w}_h^t = (\Lambda_h^t)^{-1} \sum_{i \in [t]} \sum_{m \in [M]} \phi(s_{m,h}^i, a_{m,h}^i)(R_{m,h}^i + \hat{V}_{h+1}^t(s_{m,h+1}^i)).$$

Let $\bar{G}_h^t$ be the output of R-UC-Cov-Estimation in Line 21 of Algorithm 5, and let

$$\hat{G}_h^t = \frac{1}{M} \sum_{i \in [t]} \sum_{m \in [M]} \phi(s_{m,h}^i, a_{m,h}^i) \phi(s_{m,h}^i, a_{m,h}^i)^T,$$

noting that $\hat{G}_h^t$ is simply an "unrounded" version of $\bar{G}_h^t$. That is, $\hat{G}_h^t$ would be the output of R-UC-Cov-Estimation in Line 21 if the rounding step of the algorithm was omitted. Similarly, let $\bar{\Lambda}_h^t = \bar{G}_h^t + \lambda I$ be as in Line 22 of Algorithm 5 and let $\hat{\Lambda}_h^t = \hat{G}_h^t + \lambda I$ be the "unrounded" version of $\bar{\Lambda}_h^t$.

To compress notation for partial trajectories and transitions, we write $s' \sim P_{m,h}^t$ to indicate $s' \sim P(\cdot | s_{m,h}^t, a_{m,h}^t)$, and write $\tau \sim P_h^t(\cdot | s)$ to denote sampling a partial trajectory by executing $\hat{\pi}^t$ for the remainder of the episode, starting from state $s$ at time $h$.

**Lemma D.1** (Bounding inter-policy value differences). *Let $\delta > 0$ and $\beta \in \tilde{O}(dH)$ (hiding logarithmic dependence on $1/\delta$). Let $\varepsilon = \|\bar{\mathbf{w}}_h^t - \mathbf{w}_h^t\|$ be the Euclidian distance between the rounded ridge solution output by R-LSVI-UCB and $\mathbf{w}_h^t$. Then except with probability $\delta$, for all $s \in \mathcal{S}, a \in \mathcal{A}, h \in [H], t \in [T]$,*

$$|\langle \phi(s,a), \bar{\mathbf{w}}_h^t \rangle - Q_h^\pi(s,a) - \mathbb{E}_{s' \sim P(\cdot, s, a)} [\hat{V}_{h+1}^t(s') - V_{h+1}^\pi(s')]| \leq \beta \|\phi(s,a)\|_{(\Lambda_h^t)^{-1}} + O(\varepsilon)$$

*Proof.* We first observe that

$$\langle \phi(s,a), \bar{\mathbf{w}}_h^t \rangle = \langle \phi(s,a), \bar{\mathbf{w}}_h^t - \mathbf{w}_h^t \rangle + \langle \phi(s,a), \mathbf{w}_h^t \rangle \leq \varepsilon + \langle \phi(s,a), \mathbf{w}_h^t \rangle,$$

so we will only need to show that

$$|\langle \phi(s,a), \mathbf{w}_h^t \rangle - Q_h^\pi(s,a) - \mathbb{E}_{s' \sim P(\cdot, s, a)} [\hat{V}_{h+1}^t(s') - V_{h+1}^\pi(s')]| \leq \beta \|\phi(s,a)\|_{(\Lambda_h^t)^{-1}} + O(\varepsilon).$$

By assumption, for any policy $\pi$, we can write $Q_h^\pi(\cdot, \cdot) = \langle \phi(\cdot, \cdot), \mathbf{w}_h^\pi \rangle$. It follows that for any $\pi$ we have

$$\Lambda_h^t(\mathbf{w}_h^t - \mathbf{w}_h^\pi) = \sum_{i \in [t]} \sum_{m \in [M]} \phi(s_{m,h}^i, a_{m,h}^i)(R_{m,h}^i + \hat{V}_{h+1}^t(s_{m,h+1}^i)) - \Lambda_h^t \mathbf{w}_h^\pi$$

$$= \sum_{i \in [t]} \sum_{m \in [M]} \phi(s_{m,h}^i, a_{m,h}^i)(R_{m,h}^i + \hat{V}_{h+1}^t(s_{m,h+1}^i))$$

$$- \sum_{i \in [t]} \sum_{m \in [M]} \phi(s_{m,h}^i, a_{m,h}^i)(R_{m,h}^i + \mathbb{E}_{s' \sim P_{m,h}^i} [V_{h+1}^\pi(s')]) - \lambda \mathbf{w}_h^\pi$$

$$= \sum_{i \in [t]} \sum_{m \in [M]} \phi(s_{m,h}^i, a_{m,h}^i)(\hat{V}_{h+1}^t(s_{m,h+1}^i) - \mathbb{E}_{s' \sim P_{m,h}^i} [V_{h+1}^\pi(s')]) - \lambda \mathbf{w}_h^\pi$$

$$= \sum_{i \in [t]} \sum_{m \in [M]} \phi(s_{m,h}^i, a_{m,h}^i)(\hat{V}_{h+1}^t(s_{m,h+1}^i)$$

$$+ \mathbb{E}_{s' \sim P_{m,h}^i} [\hat{V}_{h+1}^t(s') - \hat{V}_{h+1}^t(s') - V_{h+1}^\pi(s')]) - \lambda \mathbf{w}_h^\pi$$

To bound $\langle \phi(s,a), \mathbf{w}_h^t - \mathbf{w}_h^\pi \rangle$, we will separately bound

1. $\langle \phi(s,a), \lambda(\Lambda_h^t)^{-1}\mathbf{w}_h^\pi\rangle$
2. $\langle \phi(s,a), (\Lambda_h^t)^{-1}\sum_{i\in[t]}\sum_{m\in[M]}\phi(s_{m,h}^i, a_{m,h}^i)(\hat{V}_{h+1}^t(s_{m,h+1}^i) - \mathbb{E}_{s'\sim P_{m,h}^i}[\hat{V}_{h+1}^t(s')])\rangle$
3. $\langle \phi(s,a), (\Lambda_h^t)^{-1}\sum_{i\in[t]}\sum_{m\in[M]}\phi(s_{m,h}^i, a_{m,h}^i)(\mathbb{E}_{s'\sim P_{m,h}^i}[\hat{V}_{h+1}^t(s') - V_{h+1}^\pi(s')])\rangle$

The first term can be bounded as in Jin et al. (2020), applying Cauchy-Schwarz with the inner product $\langle\mathbf{x},\mathbf{y}\rangle = \mathbf{x}(\Lambda_h^t)^{-1}\mathbf{y}$ to obtain

$$|\langle \phi(s,a), \lambda(\Lambda_h^t)^{-1}\mathbf{w}_h^\pi\rangle| \leq \lambda\|\phi(s,a)\|_{(\Lambda_h^t)^{-1}}\|\mathbf{w}_h^\pi\|_{(\Lambda_h^t)^{-1}}$$
$$\leq \sqrt{d\lambda}H\|\phi(s,a)\|_{(\Lambda_h^t)^{-1}}.$$

using Lemma B.2 of Jin et al. (2020) to bound the norm $\|\mathbf{w}_h^\pi\| \leq 2H\sqrt{d}$.

To bound the second term, we again observe

$$|\langle \phi(s,a), (\Lambda_h^t)^{-1}\sum_{i\in[t]}\sum_{m\in[M]}\phi(s_{m,h}^i, a_{m,h}^i)(\hat{V}_{h+1}^t(s_{m,h+1}^i) - \mathbb{E}_{s'\sim P_{m,h}^i}[\hat{V}_{h+1}^t(s'))]\rangle|$$
$$\leq \|\phi(s,a)\|_{(\Lambda_h^t)^{-1}}\|\sum_{i\in[k]}\sum_{m\in[M]}\phi(s_{m,h}^i, a_{m,h}^i)(\hat{V}_{h+1}^t(s_{m,h+1}^i)$$
$$- \mathbb{E}_{s'\sim P_{m,h}^i}[\hat{V}_{h+1}^t(s')])\|_{(\Lambda_h^t)^{-1}},$$

so it suffices to bound
$\|\sum_{i\in[k]}\sum_{m\in[M]}\phi(s_{m,h}^t, a_{m,h}^t)(\hat{V}_{h+1}^t(s_{m,h+1}^t) - \mathbb{E}_{s'\sim P_{m,h}^t}[\hat{V}_{h+1}^t(s')])\|_{(\Lambda_h^t)^{-1}}.$

We again refer the reader to Jin et al. (2020), specifically in Section $D.2$ on Concentration of Self-Normalized Processes and Uniform Concentration over Value Functions.

Ignoring logarithmic dependence (on the covering number, $1/\delta$, core set size, and regularizer penalties) and choosing $\epsilon_{\text{net}} \propto \frac{H\sqrt{d\lambda}}{2k}$, this gives us that

$$|\langle \phi(s,a), (\Lambda_h^t)^{-1}\sum_{i\in[t]}\sum_{m\in[M]}\phi(s_{m,h}^i, a_{m,h}^i)(\hat{V}_{h+1}^t(s_{m,h+1}^i) - \mathbb{E}_{s'\sim P_{m,h}^i}[\hat{V}_{h+1}^t(s'))]\rangle|$$
$$\leq \sqrt{2}(H\sqrt{d} + \frac{2k\epsilon_{\text{net}}}{\sqrt{\lambda}})\|\phi(s,a)\|_{(\Lambda_h^t)^{-1}}$$

Finally, we bound the third term by rewriting

$$(\Lambda_h^t)^{-1}\sum_{m\in[M]}\phi(s_{m,h}^t, a_{m,h}^t)(\mathbb{E}_{s'\sim P_{m,h}^t}[\hat{V}_{h+1}^t(s') - V_{h+1}^\pi(s')])$$
$$= (\Lambda_h^t)^{-1}\sum_{m\in[M]}\phi(s_{m,h}^t, a_{m,h}^t)\phi(s_{m,h}^t, a_{m,h}^t)^T\int\hat{V}_{h+1}^t(s') - V_{h+1}^\pi(s')d\mu_h(s')$$
$$= \int\hat{V}_h^t(s') - V_{h+1}^\pi(s')d\mu_h(s') - \lambda(\Lambda_h^t)^{-1}\int\hat{V}_{h+1}^t(s') - V_{h+1}^\pi(s')d\mu_h(s')$$

It follows that
$$\langle \phi(s,a), (\Lambda_h^t)^{-1}\sum_{m\in[M]}\phi(s_{m,h}^t, a_{m,h}^t)(\mathbb{E}_{s'\sim P_{m,h}^t}[\hat{V}_{h+1}^t(s') - V_{h+1}^\pi(s')])\rangle$$
$$= \langle \phi(s,a), \int\hat{V}_{h+1}^t(s') - V_{h+1}^\pi(s')d\mu_h(s')\rangle$$
$$- \langle \phi(s,a), \lambda(\Lambda_h^t)^{-1}\int\hat{V}_{h+1}^t(s') - V_{h+1}^\pi(s')d\mu_h(s')\rangle$$
$$= \mathbb{E}_{s'\sim P(\cdot|s,a)}[\hat{V}_{h+1}^t(s') - V_{h+1}^\pi(s')] - \langle \phi(s,a), \lambda(\Lambda_h^t)^{-1}\int\hat{V}_{h+1}^t(s') - V_{h+1}^\pi(s')d\mu_h(s')\rangle$$

Finally, we can bound the rightmost inner product by

$$|\langle \phi(s,a), \lambda(\Lambda_h^t)^{-1} \int \hat{V}_{h+1}^t(s') - V_{h+1}^\pi(s') d\mu_h(s') \rangle|$$

$$\leq \lambda \|\phi(s,a)\|_{(\Lambda_h^t)^{-1}} \| \int \hat{V}_{h+1}^t(s') - V_{h+1}^\pi(s') d\mu_h(s') \|_{(\Lambda_h^t)^{-1}}$$

$$\leq \lambda H \sqrt{d} \|\phi(s,a)\|_{(\Lambda_h^t)^{-1}}$$

Putting everything together, we have that except with probability $\delta$, for any $\pi \in \Pi$, $s \in \mathcal{S}$, $a \in \mathcal{A}$, $h \in [H]$

$$|\langle \phi(s,a), \mathbf{w}_h^t \rangle - Q_h^\pi(s,a) - \mathop{\mathbb{E}}_{s' \sim P(\cdot|s,a)} [\hat{V}_{h+1}^t(s') - V_{h+1}^\pi(s')]|$$

$$= |\langle \phi(s,a), \mathbf{w}_h^t - \mathbf{w}_h^\pi \rangle - \mathop{\mathbb{E}}_{s' \sim P(\cdot|s,a)} [\hat{V}_{h+1}^t(s') - V_{h+1}^\pi(s')]|$$

$$\leq \beta \|\phi(s,a)\|_{(\Lambda_h^t)^{-1}} + O(\varepsilon)$$

$\square$

For the purposes of proving the UCB property (Lemma D.4) and our regret bound (Lemma D.5), we will need the following lemma which bounds the Mahalanobis norm of a vector with respect to a matrix $\bar{\Lambda}$ in terms of the Mahalanobis norm of the same vector with respect to a small perturbation of $\bar{\Lambda}$.

**Lemma D.2.** *Let $\Lambda$ be a positive definite matrix with smallest eigenvalue at least $\lambda$. Let $E$ be a positive semidefinite matrix for which and $\|E\| < \varepsilon$ and let $\bar{\Lambda}$ be such that $\bar{\Lambda} = \Lambda + E$. Then for all $s \in \mathcal{S}$, $a \in \mathcal{A}$,*

$$|\|\phi(s,a)\|_{\bar{\Lambda}^{-1}} - \|\phi(s,a)\|_{\Lambda^{-1}}| \leq \sqrt{\frac{\varepsilon}{\lambda - \varepsilon}}$$

*Proof.* It follows from the Neumann series for $(1 + E)^{-1}$ that

$$\bar{\Lambda}^{-1} = (\Lambda + E)^{-1}$$

$$= \Lambda^{-1}(I + E(\Lambda)^{-1})^{-1}$$

$$= \Lambda^{-1} \sum_{i=0}^{\infty} (-E\Lambda^{-1})^i$$

$$= \Lambda^{-1} + \sum_{i=1}^{\infty} (-E\Lambda^{-1})^i$$

and so

$$\phi(s,a)^T \bar{\Lambda}^{-1} \phi(s,a) = \phi(s,a)^T (\Lambda^{-1} + \sum_{i=1}^{\infty} (-E\Lambda^{-1})^i) \phi(s,a)$$

$$= \phi(s,a)^T \Lambda^{-1} \phi(s,a) + \phi(s,a)^T \sum_{i=1}^{\infty} (-E\Lambda^{-1})^i \phi(s,a).$$

It follows that

$$|\phi(s,a)^T \bar{\Lambda}^{-1} \phi(s,a) - \phi(s,a)^T \Lambda^{-1} \phi(s,a)| \leq \sum_{i=1}^{\infty} \|(E\Lambda^{-1})^i\|$$

$$\leq \sum_{i=1}^{\infty} \|E\|^i \|\Lambda^{-1}\|^i$$

$$\leq \sum_{i=1}^{\infty} \lambda^{-i} \|E\|^i$$

$$\leq \frac{\|E\|}{\lambda - \|E\|}$$

$$\leq \frac{\varepsilon}{\lambda - \varepsilon}$$

Using the fact that for non-negative $a, b$ if $|a - b| \leq c$, then $|\sqrt{a} - \sqrt{b}| \leq \sqrt{c}$. This also implies that

$$|\|\phi(s,a)\|_{(\bar{\Lambda}_h^t)^{-1}} - \|\phi(s,a)\|_{(\Lambda_h^t)^{-1}}| \leq \sqrt{\frac{\varepsilon}{\lambda - \varepsilon}}$$

$\square$

**Lemma D.3.** *So long as the rounding error incurred by* R-UC-Cov-Estimation *satisfies* $\|\bar{\Lambda}_h^t - \hat{\Lambda}_h^t\| \leq \frac{\lambda \varepsilon^2}{2\beta^2 H^2}$, *then for all* $s \in \mathcal{S}, a \in \mathcal{A}, h \in [H], k \in [K]$, *it holds that*

$$\|\phi(s,a)\|_{(\bar{\Lambda}_h^t)^{-1}} \geq \|\phi(s,a)\|_{(\Lambda_h^t)^{-1}} - \frac{\varepsilon}{\beta H}$$

*and*

$$\|\phi(s,a)\|_{(\bar{\Lambda}_h^t)^{-1}} \leq \|\phi(s,a)\|_{(\hat{\Lambda}_h^t)^{-1}} + \frac{\varepsilon}{\beta H}$$

*Proof.* Recalling that

$$\hat{\Lambda}_h^t = \frac{1}{M} \sum_{i \in [t]} \sum_{m \in [M]} \phi(s_{m,h}^i, a_{m,h}^i) \phi(s_{m,h}^i, a_{m,h}^i)^T + \lambda I$$

and

$$\Lambda_h^t = \sum_{i \in [t]} \sum_{m \in [M]} \phi(s_{m,h}^i, a_{m,h}^i) \phi(s_{m,h}^i, a_{m,h}^i)^T + \lambda I$$

it immediately follows that

$$\phi(s,a) \hat{\Lambda}_h^t \phi(s,a) \leq \phi(s,a) \Lambda_h^t \phi(s,a)$$

and therefore

$$\|\phi(s,a)\|_{(\hat{\Lambda}_h^t)^{-1}} \geq \|\phi(s,a)\|_{(\Lambda_h^t)^{-1}}$$

By Lemma D.2, we have that

$$|\|\phi(s,a)\|_{(\bar{\Lambda}_h^t)^{-1}} - \|\phi(s,a)\|_{(\hat{\Lambda}_h^t)^{-1}}| \leq \sqrt{\frac{\|E\|}{\lambda - \|E\|}}$$

where $E = \bar{\Lambda}_h^t - \hat{\Lambda}_h^t$. Then so long as $\|E\| \leq \frac{\lambda \varepsilon^2}{2\beta^2 H^2}$, we have that $\sqrt{\frac{\|E\|}{\lambda - \|E\|}} \leq \frac{\epsilon}{\beta H}$ $\square$

We now prove that the UCB property holds for Algorithm 5. That is, the predicted value of any state-action pair for the current policy is always greater than the true value of that state-action pair under any alternative policy, up to some small, controllable estimation error.

**Lemma D.4** (Upper-confidence bound). *Let* $\delta > 0$ *and* $\beta \in \tilde{O}(dH)$ *(hiding logarithmic dependence on* $1/\delta$*). Let* $\|\bar{\mathbf{w}}_h^t - \mathbf{w}_h^t\| \leq \frac{\varepsilon}{H}$ *be the Euclidian distance between the rounded ridge solution output by* R-LSVI-UCB *and* $\mathbf{w}_h^t$. *Assume that for all* $s \in \mathcal{S}, a \in \mathcal{A}, h \in [H], k \in [K]$, *it also holds that*

$$\|\phi(s,a)\|_{(\bar{\Lambda}_h^t)^{-1}} \geq \|\phi(s,a)\|_{(\Lambda_h^t)^{-1}} - \frac{\varepsilon}{\beta H}$$

*Then except with probability* $\delta$*, for all* $s \in \mathcal{S}, a \in \mathcal{A}, h \in [H], k \in [K]$*, and all* $\pi \in \Pi$:

$$\hat{Q}_h^t(s,a) \geq Q_h^\pi(s,a) - O(\varepsilon)$$

*Proof.* We will prove the lemma by induction on $h$. Assume that

$$\hat{Q}^t_{h+1}(s,a) \geq Q^\pi_{h+1}(s,a) - (H-h)O(\tfrac{\varepsilon}{H}).$$

Then we can use Lemma D.1 to argue

$$
\begin{aligned}
\hat{Q}^t_h(s,a) &= \min\{H, \langle \phi(s,a), \bar{\mathbf{w}}^t_h \rangle + \beta\|\phi(s,a)\|_{(\bar{\Lambda}^t_h)^{-1}}\} \\
&\geq \min\{H, \langle \phi(s,a), \mathbf{w}^t_h \rangle + \beta\|\phi(s,a)\|_{(\Lambda^t_h)^{-1}} - O(\tfrac{\varepsilon}{H})\} \\
&\geq \min\{H, Q^\pi_h(s,a) + \underset{s'\sim P(\cdot|s,a)}{\mathbb{E}}[\hat{V}^t_{h+1}(s') - V^\pi_{h+1}(s')] - O(\tfrac{\varepsilon}{H})\} \quad \text{by Lemma D.1} \\
&\geq Q^\pi_h(s,a) - (H - h + 1)O(\tfrac{\varepsilon}{H}) \qquad\qquad\qquad\qquad\qquad \text{by induction}
\end{aligned}
$$

To see that the base case holds for $h = H - 1$, we observe that from Lemma D.1,

$$|\langle \phi(s,a), \bar{\mathbf{w}}^t_{H-1} \rangle - Q^\pi_{H-1}(s,a)| \leq \beta\|\phi(s,a)\|_{(\Lambda^t_{H-1})^{-1}} + O(\tfrac{\varepsilon}{H})$$

and therefore $Q^\pi_{H-1}(s,a) \leq \langle \phi(s,a), \bar{\mathbf{w}}^t_{H-1} \rangle + \beta\|\phi(s,a)\|_{(\Lambda^t_{H-1})^{-1}} + O(\tfrac{\varepsilon}{H})$. We defined $\hat{Q}^t_{H-1}(s,a) = \langle \phi(s,a), \bar{\mathbf{w}}^t_{H-1} \rangle + \beta\|\phi(s,a)\|_{(\Lambda^t_{H-1})^{-1}}$, and so it follows that

$$\hat{Q}^t_{H-1}(s,a) \geq Q^\pi_h(s,a) - O(\tfrac{\varepsilon}{H})$$

$\square$

In order to bound the contribution of the UCB bonus term to the regret of our learner, we first bound the sum of the Mahalanobis norms

$$\sum_{t\in[T]}\sum_{h\in[H]}\sum_{m\in[M]} \|\phi(s^{t+1}_{m,h}, a^{t+1}_{m,h})\|_{(\hat{G}^t_h)^{-1}},$$

under the "unrounded" matrices $\hat{G}^t_h$. We then use Lemma D.2 to show that the rounding to ensure replicability does not increase the overall regret too much, obtaining a bound on

$$\sum_{t\in[T]}\sum_{h\in[H]}\sum_{m\in[M]} \|\phi(s^{t+1}_{m,h}, a^{t+1}_{m,h})\|_{(\bar{G}^t_h)^{-1}},$$

the actual contribution to the regret from the bonus term.

**Lemma D.5** (UCB regret).

$$\sum_{t\in[T]}\sum_{h\in[H]}\sum_{m\in[M]} \|\phi(s^{t+1}_{m,h}, a^{t+1}_{m,h})\|_{(\hat{G}^t_h)^{-1}} \leq MH\sqrt{T(1+1/\lambda)d\log\left(\frac{\lambda+T}{\lambda}\right)}$$

*Proof.* We observe that for any $t \in [T]$, $h \in [H]$, $s \in \mathcal{S}$, $a \in \mathcal{A}$, that because $\lambda_{\min}(\hat{G}^t_h) \geq \lambda$,

$$\|\phi(s,a)\|^2_{(\hat{G}^t_h)^{-1}} \leq \tfrac{1}{\lambda_{\min}(\hat{G}^t_h)}\|\phi(s,a)\|^2 \leq \tfrac{1}{\lambda}.$$

It follows from $\ln(1 + x) \leq x \leq (1+x)\ln(1+x)$ for all $x \geq -1$ then, that

$$
\begin{aligned}
\ln(1 + \|\phi(s,a)\|^2_{(\hat{G}^t_h)^{-1}}) &\leq \|\phi(s,a)\|^2_{(\hat{G}^t_h)^{-1}} \\
&\leq (1+1/\lambda)\ln(1 + \|\phi(s,a)\|^2_{(\hat{G}^{t-1}_h)^{-1}})
\end{aligned}
$$

From the definition of $\hat{G}^t_h$, the matrix determinant lemma, and the fact that $\det(I + X) \geq 1 + \operatorname{tr}(X)$ for positive semidefinite $X$, we have that

$$
\begin{aligned}
\det(\hat{G}^{t+1}_h) &= \det(\lambda I + \frac{1}{M}\sum_{i\in[t+1]}\sum_{m\in[M]} \phi(s^i_{m,h}, a^i_{m,h})\phi(s^i_{m,h}, a^i_{m,h})^T) \\
&= \det(\hat{G}^t_h + \frac{1}{M}\sum_{m\in[M]} \phi(s^t_{m,h}, a^t_{m,h})\phi(s^t_{m,h}, a^t_{m,h})^T)
\end{aligned}
$$

$$= \det(\hat{G}_h^t) \det(I + \tfrac{1}{M}(\hat{G}_h^t)^{-1} \sum_{m \in [M]} \phi(s_{m,h}^t, a_{m,h}^t)\phi(s_{m,h}^t, a_{m,h}^t)^T)$$

$$\geq \det(\hat{G}_h^t)(1 + \tfrac{1}{M} \sum_{m \in [M]} \|\phi(s_{m,h}^t, a_{m,h}^t)\|_{(\hat{G}_h^t)^{-1}}^2)$$

It follows that

$$\ln(1 + \tfrac{1}{M} \sum_{m \in [M]} \|\phi(s_{m,h}^t, a_{m,h}^t)\|_{(\hat{G}_h^t)^{-1}}^2) \leq \ln \frac{\det(\hat{G}_h^{t+1})}{\det(\hat{G}_h^t)}.$$

Then

$$\tfrac{1}{M} \sum_{t \in [T]} \sum_{h \in [H]} \sum_{m \in [M]} \|\phi(s_{m,h}^{t+1}, a_{m,h}^{t+1})\|_{(\hat{G}_h^t)^{-1}}^2$$

$$\leq (1 + 1/\lambda) \sum_{t \in [T]} \sum_{h \in [H]} \ln(1 + \tfrac{1}{M} \sum_{m \in [M]} \|\phi(s_{m,h}^{t+1}, a_{m,h}^{t+1})\|_{(\hat{G}_h^t)^{-1}}^2)$$

$$\leq (1 + 1/\lambda) \sum_{t \in [T]} \sum_{h \in [H]} \ln \frac{\det(\hat{G}_h^{t+1})}{\det(\hat{G}_h^t)}$$

$$= (1 + 1/\lambda) \sum_{h \in [H]} \ln \left( \frac{\det(\hat{G}_h^T)}{\det(\hat{G}_h^0)} \right)$$

$$\leq (1 + 1/\lambda) \sum_{h \in [H]} \ln \left( \frac{(\lambda + T)^d}{\det(\hat{G}_h^0)} \right)$$

$$= (1 + 1/\lambda) H d \ln \left( \frac{\lambda + T}{\lambda} \right)$$

where the third inequality follows from the fact that

$$\det(\hat{G}_h^T) \leq (\lambda + \tfrac{1}{Md} \sum_{i \in [T]} \sum_{m \in [M]} \|\phi(s_{m,h}^i, a_{m,h}^i)\|_2^2)^d \leq (\lambda + T)^d.$$

Then applying Cauchy-Schwarz to bound the actual sums of the norms, rather than the quadratic forms, we have

$$\sum_{t \in [T]} \sum_{m \in [M]} \sum_{h \in [H]} \|\phi(s_{m,h}^{t+1}, a_{m,h}^{t+1})\|_{(\hat{G}_h^t)^{-1}} \leq MH \sqrt{T(1 + 1/\lambda)d \log \left( \frac{\lambda + T}{\lambda} \right)}$$

$$\square$$

Using Lemma D.5 and Lemma D.3, we bound the real contribution of the bonus term to the overall regret, accounting for the error induced by rounding for replicability.

**Corollary D.1.**

$$\sum_{t \in [T]} \sum_{h \in [H]} \sum_{m \in [M]} \|\phi(s_{m,h}^{t+1}, a_{m,h}^{t+1})\|_{(\bar{G}_h^t)^{-1}} \leq 2MH \sqrt{T(1 + 1/\lambda)d \log \left( \frac{\lambda + T}{\lambda} \right)}$$

*Proof.* Lemma D.2 gives us

$$\sum_{t \in [T]} \sum_{h \in [H]} \sum_{m \in [M]} \|\phi(s_{m,h}^{t+1}, a_{m,h}^{t+1})\|_{(\bar{G}_h^t)^{-1}} \leq \sum_{t \in [T]} \sum_{m \in [M]} \sum_{h \in [H]} \|\phi(s_{m,h}^{t+1}, a_{m,h}^{t+1})\|_{(\hat{G}_h^t)^{-1}} + \sqrt{\frac{\|E\|}{\lambda - \|E\|}}$$

so as long as we ensure

$$\|E\| \leq \frac{\lambda d \log(\frac{\lambda + T}{\lambda})}{2T},$$

it holds that

$$\sum_{t\in[T]}\sum_{h\in[H]}\sum_{m\in[M]}\|\phi(s_{m,h}^{t+1},a_{m,h}^{t+1})\|_{(\bar{G}_h^t)^{-1}} \leq 2MH\sqrt{T(1+1/\lambda)d\log\left(\frac{\lambda+T}{\lambda}\right)}$$

$\square$

We can now use Lemma D.1 and Lemma D.4 to prove Theorem 4.2.

*Proof.* Let $\Pi^t$ denote the set of policies output at the end of the algorithm. We want to show that, except with probability $\delta$, for all $\pi\in\Pi$

$$\mathbb{E}_{t\sim[T]}[V_q^t] \geq V_q^\pi - O(\varepsilon).$$

Assume in the following that for all $t\in[T], h\in[H]$ we have $\|\mathbf{w}_h^t - \bar{\mathbf{w}}_h^t\| \leq \Delta_w$ and for all $s\in\mathcal{S}$, $a\in\mathcal{A}$, we have $\|\phi(s,a)\|_{(\hat{\Lambda}_h^t)^{-1}} - \|\phi(s,a)\|_{(\bar{\Lambda}_h^t)^{-1}} \leq \Delta_\Lambda$.

From Lemma D.4, we have that for every $t\in[T], V_q^\pi - V_q^t \leq \hat{V}_q^t - V_q^t + O(H\Delta_w)$, and so it will be enough to bound

$$\frac{1}{T}\sum_{t\in[T]}[\hat{V}_q^t - V_q^t] = \frac{1}{T}\sum_{t\in[T]}\mathbb{E}_{s_0\sim q}[\hat{Q}_0^t(s_0,\hat{\pi}_0^t(s_0)) - Q_0^t(s_0,\hat{\pi}_0^t(s_0))]$$

$$= \frac{1}{TM}\sum_{t\in[T]}\sum_{m\in[M]}\hat{Q}_0^t(s_{m,0}^t,a_{m,0}^t) - Q_0^t(s_{m,0}^t,a_{m,0}^t) + err$$

where

$$err = \sum_{t\in[T]}\sum_{m\in[M]}\mathbb{E}_{s_0\sim q}[\hat{Q}_0^t(s_0,\hat{\pi}_0^t(s_0)) - Q_0^t(s_0,\hat{\pi}_0^t(s_0))] - \hat{Q}_0^t(s_{m,0}^t,a_{m,0}^t)) - Q_0^t(s_{m,0}^t,a_{m,0}^t)$$

$$\leq H\sqrt{2TM\log(1/\delta)}$$

except with probability $\delta$.

We begin by bounding

$$\sum_{t\in[T]}\sum_{m\in[M]}\hat{Q}_0^t(s_{m,0}^t,a_{m,0}^t) - Q_0^t(s_{m,0}^t,a_{m,0}^t).$$

Applying Lemma D.1 to $\pi = \hat{\pi}^t$, we have that except with probability $\delta$

$$|\langle\phi(s,a),\bar{\mathbf{w}}_h^t\rangle - Q_h^t(s,a) - \mathbb{E}_{s'\sim P(\cdot,s,a)}[\hat{V}_{h+1}^t(s') - V_{h+1}^t(s')]| \leq \beta\|\phi(s,a)\|_{(\Lambda_h^t)^{-1}} + O(\Delta_w)$$

and therefore

$$\hat{Q}_h^t(s_{m,h}^t,a_{m,h}^t) - Q_h^t(s_{m,h}^t,a_{m,h}^t) \leq \mathbb{E}_{s'\sim P_{m,h}^t}[\hat{V}_{h+1}^t(s') - V_{h+1}^t(s')] + \beta\|\phi(s_{m,h}^t,a_{m,h}^t)\|_{(\bar{\Lambda}_h^t)^{-1}}$$

$$+ \beta\|\phi(s_{m,h}^t,a_{m,h}^t)\|_{(\Lambda_h^t)^{-1}} + O(\Delta_w)$$

$$= \hat{V}_{h+1}^t(s_{m,h+1}^t) - V_{h+1}^t(s_{m,h+1}^t) + \mathbb{E}_{s'\sim P_{m,h}^t}[\hat{V}_{h+1}^t(s') - V_{h+1}^t(s')]$$

$$- (\hat{V}_{h+1}^t(s_{m,h+1}^t) - V_{h+1}^t(s_{m,h+1}^t)) + \beta\|\phi(s_{m,h}^t,a_{m,h}^t)\|_{(\Lambda_h^t)^{-1}}$$

$$+ \beta\|\phi(s_{m,h}^t,a_{m,h}^t)\|_{(\bar{\Lambda}_h^t)^{-1}} + O(\Delta_w)$$

which gives

$$\sum_{t\in[T]}\sum_{m\in[M]}\hat{Q}_0^t(s_{m,0}^t,a_{m,0}^t) - Q_0^t(s_{m,0}^t,a_{m,0}^t)$$

$$\leq \sum_{t\in[T]} \sum_{m\in[M]} \sum_{h\in[H]} \mathop{\mathbb{E}}_{s'\sim P^t_{m,h}} [\hat{V}^t_{h+1}(s') - V^t_{h+1}(s')] - (\hat{V}^t_{h+1}(s^t_{m,h+1}) - V^t_{h+1}(s^t_{m,h+1}))$$

$$+ \sum_{t\in[T]} \sum_{m\in[M]} \sum_{h\in[H]} \beta\|\phi(s^t_{m,h}, a^t_{m,h})\|_{(\Lambda^t_h)^{-1}}$$

$$+ \beta\|\phi(s^t_{m,h}, a^t_{m,h})\|_{(\bar{\Lambda}^t_h)^{-1}} + O(\Delta_w)$$

$$\leq \sum_{t\in[T]} \sum_{m\in[M]} \sum_{h\in[H]} \mathop{\mathbb{E}}_{s'\sim P^t_{m,h}} [\hat{V}^t_{h+1}(s') - V^t_{h+1}(s')] - (\hat{V}^t_{h+1}(s^t_{m,h+1}) - V^t_{h+1}(s^t_{m,h+1}))$$

$$+ \sum_{t\in[T]} \sum_{m\in[M]} \sum_{h\in[H]} 2\beta\|\phi(s^t_{m,h}, a^t_{m,h})\|_{(\bar{\Lambda}^t_h)^{-1}} + O(\tfrac{\beta\sqrt{\Delta_\Lambda}}{\lambda}) + O(\Delta_w),$$

where the last inequality follows from Lemma D.3. From Lemma D.5, it follows that

$$\sum_{t\in[T]} \sum_{m\in[M]} \hat{Q}^t_0(s^t_{m,0}, a^t_{m,0}) - Q^t_0(s^t_{m,0}, a^t_{m,0})$$

$$\leq \sum_{t\in[T]} \sum_{m\in[M]} \sum_{h\in[H]} \underbrace{\mathop{\mathbb{E}}_{s'\sim P^t_{m,h}} [\hat{V}^t_{h+1}(s') - V^t_{h+1}(s')] - (\hat{V}^t_{h+1}(s^t_{m,h+1}) - V^t_{h+1}(s^t_{m,h+1}))}_{\alpha^t_{m,h}}$$

$$+ 4\beta M H \sqrt{T(1 + 1/\lambda)d \log\left(\frac{\lambda + T}{\lambda}\right)} + O(\tfrac{\beta\sqrt{\Delta_\Lambda}}{\lambda}) + O(\Delta_w)$$

It remains to bound $\sum_{t\in[T]} \sum_{m\in[M]} \sum_{h\in[H]} \alpha^t_{m,h}$. We observe that the sum of $\alpha^t_{m,h}$ is the sum of deviations of $\hat{V}^t_{h+1}(s^t_{m,h+1}) - V^t_{h+1}(s^t_{m,h+1})$ from their expectation $\mathbb{E}_{s'\sim P^t_{m,h}}[\hat{V}^t_{h+1}(s') - V^t_{h+1}(s')]$ and represents a Martingale sequence with every $|\alpha^t_{m,h}| \leq 2H$ bounded r.v.'s. Therefore

$$\mathbf{Pr}[\sum_{t\in[T]} \sum_{m\in[M]} \sum_{h\in[H]} \alpha^t_{m,h} > \tau] \leq \exp(\tfrac{-\tau^2}{2TMH^3})$$

and then

$$\mathbf{Pr}[\sum_{t\in[T]} \sum_{m\in[M]} \sum_{h\in[H]} \alpha^t_{m,h} > H\sqrt{2\log(1/\delta)TMH}] \leq \delta$$

Putting everything together, we have that

$$\tfrac{1}{T} \sum_{t\in[T]} [\hat{V}^t_q - V^t_q] = \tfrac{1}{TM} \sum_{t\in[T]} \sum_{m\in[M]} \hat{Q}^t_0(s^t_{m,0}, a^t_{m,0}) - Q^t_0(s^t_{m,0}, a^t_{m,0}) + err$$

$$\leq \tfrac{1}{TM}(H\sqrt{2\log(1/\delta)TMH} + 4\beta M H \sqrt{T(1 + 1/\lambda)d \log(\tfrac{\lambda+T}{\lambda})}$$

$$+ TMHO(\tfrac{\beta\sqrt{\Delta_\Lambda}}{\lambda} + \Delta_w) + H\sqrt{2TM\log(1/\delta)})$$

$$= \sqrt{\tfrac{2H^3\log(1/\delta)}{TM}} + \tfrac{4\beta H}{\sqrt{T}}\sqrt{(1 + 1/\lambda)d\log(\tfrac{\lambda+T}{\lambda})} + HO(\tfrac{\beta\sqrt{\Delta_\Lambda}}{\lambda} + \Delta_w) + H\sqrt{\tfrac{2\log(1/\delta)}{TM}}$$

except with probability $2\delta$.

Then taking $T \in \tilde{O}\left(\frac{H^3\log(1/\delta)}{M\varepsilon^2} + \frac{\beta^2 H^2 d}{\lambda\varepsilon^2}\right) \in \tilde{O}\left(\frac{\beta^2 H^2 d\log(1/\delta)}{\lambda\varepsilon^2}\right)$ and ensuring $\Delta_\Lambda \in O((\frac{\varepsilon\lambda}{H\beta})^2)$ and $\Delta_w \in O(\frac{\varepsilon}{H})$, we have that

$$\tfrac{1}{T} \sum_{t\in[T]} [\hat{V}^t_q - V^t_q] \leq \varepsilon$$

except with probability $\delta$. $\qquad\square$

## D.2 REPLICABILITY PROOF FOR THEOREM 4.2

*Proof.* We prove that Alg. 5 is $\rho$-replicable by strong induction over rounds. Let $\hat{\pi}^{t,(1)}$ and $\hat{\pi}^{t,(2)}$ be the policies returned at the end of round $t$ by two independent executions that share the same internal randomness.

Base case ($t = 0$): Initialization is deterministic, hence $\hat{\pi}^{0,(1)} = \hat{\pi}^{0,(2)}$ with probability 1.

Inductive step: For $k \in \{0, \ldots, T-1\}$, assume $\mathcal{E}_k := \{\hat{\pi}^{i,(1)} = \hat{\pi}^{i,(2)} \, \forall i \leq k\}$ holds. Conditioned on $\mathcal{E}_k$, both runs collect data in round $k+1$ using the same mixture over the policies $\{\hat{\pi}^0, \ldots, \hat{\pi}^k\}$ (the mixture indices are fixed by the shared internal randomness). For each step $h \in [H]$, this induces the same distribution $D_h^{[k]}$ over design/label pairs $(\phi(s_h, a_h), y_h)$ in both runs, where $y_h$ is the reward-plus-value target with the rounded bonus. Within each round, the $M$ trajectories (and their step-$h$ samples) are i.i.d.; across rounds, fresh trajectories are drawn, so data blocks are independent once the policy sequence is fixed by $r$ (i.e., under $\mathcal{E}_k$).

By Theorem 3.1, with per-call parameter $\rho_{\mathrm{rdg}}$ and target $\Delta_w$ and with $M$ large enough as specified there, the rounded ridge outputs coincide: $\bar{\mathbf{w}}_h^{k+1,(1)} = \bar{\mathbf{w}}_h^{k+1,(2)}$ except with probability at most $\rho_{\mathrm{rdg}}$. By Theorem 3.3, with per-call parameter $\rho_\Lambda$ and target $\Delta_\Lambda$ and with the corresponding $M$, the rounded covariances also coincide: $\bar{G}_h^{k+1,(1)} = \bar{G}_h^{k+1,(2)}$ (hence $\bar{\Lambda}_h^{k+1}$ coincide) except with probability at most $\rho_\Lambda$. The algorithm sets $\rho_{\mathrm{rdg}} = \rho_\Lambda = \rho/(4HK)$ and chooses $M$ to satisfy the sample-size requirements of Theorems 3.1 and 3.3 for the targets $\Delta_w$ and $\Delta_\Lambda$ used in the accuracy analysis.

When both estimators succeed for all $h \in [H]$, the Q-estimates and bonuses are identical at every state-action pair, and the greedy action selection with deterministic tie-breaking yields the same $\hat{\pi}^{k+1}$ in both runs. Taking $\rho_{\mathrm{rdg}} = \rho_\Lambda = \rho/(4HK)$ and union-bounding over the $2H$ estimator calls in round $k+1$ shows

$$\mathbf{Pr}[\hat{\pi}^{k+1,(1)} \neq \hat{\pi}^{k+1,(2)} \mid \mathcal{E}_k] \leq \tfrac{\rho}{K}.$$

Unconditioning and using the inductive hypothesis yields

$$\mathbf{Pr}[\hat{\pi}^{k+1,(1)} \neq \hat{\pi}^{k+1,(2)}] \leq \mathbf{Pr}[\neg \mathcal{E}_k] + \tfrac{\rho}{K} \leq \tfrac{k\rho}{K} + \tfrac{\rho}{K} = \tfrac{(k+1)\rho}{K}.$$

By induction, after $T$ rounds we have $\mathbf{Pr}[\hat{\pi}^{T,(1)} \neq \hat{\pi}^{T,(2)}] \leq \rho$, i.e., Alg. 5 is $\rho$-replicable. Finally, each round uses fresh trajectories; conditioned on $\mathcal{E}_k$, the batch at round $k+1$ is i.i.d. from $D_h^{[k]}$, so the prerequisites of Theorems 3.1 and 3.3 hold at every call. $\qquad\square$

## E HYPERPARAMETERS

For our neural network experiments, we use the implementation of PQN available via CleanRL (Huang et al., 2022). We report the hyperparameters that we used in Table 1. We found that from the original implementation we need to make minor modifications that do not lead to decrease in performance of the baseline to get competitive performance with quantization. Precisely, we change the default exploration rate from $0.1$ to $0.25$ for MsPacman and to $0.3$ for Breakout because quantization leads to lower policy churn which has been shown to increase exploration (Schaul et al., 2022). This is an intended side effect. While one might be tempted to argue that exploration from churn is good, it is uncontrolled as we do not know how our networks change. Instead, quantization leads to lower churn and the modeler can control the rate of exploration directly. Furthermore, we change the optimizer to use decoupled weight decay (Loshchilov & Hutter, 2019). We use a weight decay of $0.1$ for MsPacman and $0.2$ for Breakout. All outputs are rounded to a multiple of $0.4$ during quantization.

Table 1: Hyperparameters for PQN

| optimizer | AdamW |
|---|---|
| total_timesteps | $10e6$ |
| learning_rate | $2.5e-4$ |
| num_envs | 8 |
| num_steps | 128 |
| anneal_lr | True |
| $\gamma$ | 0.99 |
| num_mini_batches | 4 |
| update_epochs | 4 |
| max_grad_norm | 10.0 |
| start_e | 1.0 |
| end_e | 0.01 |
| exploration_fraction | 0.3 |
| q_lambda | 0.65 |

## F  COMPUTATIONAL RESOURCES

Our code is written in Python and uses PyTorch for deep learning. Our algorithms for CartPole can be run on household-grade computers using using central processing units (CPUs). For deep learning experiments, we had access to a cluster with various types of graphical processing units (GPUs), including Nvidia RTX3090 and Nvidia A6000 GPUs. Running one seed of the fitted Q-iteration algorithm less than a minute while running one PQN experiment takes around 2.5 hours per seed.

## G  LARGE LANGUAGE MODEL USAGE

During the development of the paper, we made use of Large Language Models (LLMs) in two ways. First, we used it to obtain sentence or word suggestions to polish text. Second, we used LLMs as a search engine to find related literature and work to specific theorem statements.

