# OpenReview forum: "Replicable Reinforcement Learning with Linear Function Approximation"
_ICLR.cc/2026/Conference — ICLR 2026 Poster_

### Official Review · Reviewer_2UsG · 2025-10-26

**Soundness:** 4
**Presentation:** 4
**Contribution:** 4
**Rating:** 8
**Confidence:** 5

**Summary:**

This paper addresses the critical challenge of replicability in reinforcement learning, where algorithms are notoriously unstable, especially when combined with function approximation[lines: 1243, 1253, 1254].The authors extend the formal, "exact" notion of replicability—where an algorithm must produce identical outputs from different data samples given the same internal randomness [lines: 1403, 1404]from the tabular setting [lines: 1246, 1270] to the more practical domain of linear function approximation[lines: 1247, 1323].

To achieve this, the paper makes several key contributions:
1. It first develops two novel, provably replicable statistical tools: **R-Ridge-Regression** for the random design setting [lines: 1248, 1516] and **R-UC-Cov-Estimation** for uncentered covariance matrices[lines: 1248, 1616]. Both are of independent interest and are built upon randomized rounding techniques [lines: 1414, 1516, 1618].
2.  Using these tools, the paper presents the **first provably efficient and replicable RL algorithms for linear MDPs**. This includes **R-LSVI** for the generative model setting [lines: 1249, 1647] and **R-LSVI-UCB** for the more challenging episodic exploration setting [lines: 1249, 1728].
3.  The theoretical results are supported by experiments. First, on CartPole with random Fourier features, the authors show their method achieves replicability with far fewer samples than the loose theoretical bounds suggest [lines: 1799, 1805, 1843]. Second, they apply the *idea* of their work (quantization) to deep RL (PQN on Atari) and demonstrate that it leads to significantly higher policy agreement across random seeds without compromising performance [lines: 1800, 1869, 1915, 1916].

The paper's theoretical claims are well-supported and appear sound. The authors properly adopt the formal replicability framework from Impagliazzo et al. (2022) [lines: 1401, 1403] and build upon the standard analysis of linear MDPs from Jin et al. (2020). The core of the contribution lies in the novel replicable estimators for ridge regression and covariance. The proofs for these tools correctly use properties of strong convexity and concentration bounds to show that the estimators from two different datasets will be close enough to be mapped to the same rounded value [lines: 1517, 1525, 2759-2763]. The subsequent RL algorithm proofs correctly follow by induction, carefully accounting for the error propagation from batching and rounding [lines: 3747, 3757, 3887]. The experimental methodology is clear, and the results convincingly support the paper's claims.

The paper is exceptionally well-written, clear, and logically structured. It begins with a strong motivation for the problem, clearly defines the technical preliminaries (linear MDPs and the specific replicability definition) [lines: 1331-1436], and then intelligently builds its contributions. The decision to first introduce the replicable statistical "tools" (regression and covariance) in a standalone section [line: 1444] before applying them to the more complex RL setting is highly effective for clarity. The experiments are well-described, and the figures clearly visualize the trade-offs between performance and policy agreement [lines: 1809, 1866].

The contribution is original and significant. The lack of replicability is a major, well-documented practical problem in RL [lines: 1243, 1252], and this paper is the first to provide a *formal, theoretical* solution in the non-tabular setting of linear function approximation [lines: 1247, 1324]. The new algorithms, R-Ridge-Regression (for random designs) [line: 1515] and R-UC-Cov-Estimation [line: 1616], are novel contributions in their own right and may have applications beyond RL [line: 1447]. This work successfully bridges a major gap and opens up a new and important line of research into provably stable and reliable RL.

**Strengths:**

* **Novelty and Significance:** This is the first work to provide provably replicable algorithms for RL with linear function approximation [lines: 1247, 1324]. It tackles the well-known "replication crisis" in RL [line: 1243] with theoretical rigor, moving beyond the tabular setting.
* **New Replicable Estimators:** The paper develops R-Ridge-Regression and R-UC-Cov-Estimation, two new statistical estimators with replicability guarantees [lines: 1248, 1445, 1446]. The extension of replicable regression to the *random design* setting [line: 1515] (as opposed to fixed design) is a key tool that enables the subsequent RL results.
* **Comprehensive Theoretical Results:** The paper provides a complete theoretical treatment, developing replicable algorithms for both the generative model setting (R-LSVI with core set) [line: 1647] and the more difficult episodic exploration setting (R-LSVI-UCB) [line: 1728].
* **Strong Experimental Validation:** The authors provide two sets of experiments:
    1.  A direct validation of the theory on CartPole, which encouragingly shows that replicability is achieved with far fewer samples than the theoretical bounds require [lines: 1804, 1843].
    2.  A practical extension to deep RL (Atari), showing the core idea of quantization inspired by the theory leads to measurably more consistent policies (higher action agreement) without a loss in performance [lines: 1800, 1915, 1916].

**Weaknesses:**

While this is a strong paper, its weaknesses are primarily the necessary trade-offs for achieving such a strong replicability guarantee.

1.  **High Sample Complexity:** The most notable weakness, which the authors acknowledge, is the very high sample complexity of the provable algorithms [line: 1733]. For example, the bound for R-LSVI-UCB has high-polynomial dependencies on $d$ and $H$ (e.g., $d^{56}H^{62}$), making the *theoretical* bounds impractical, even if the empirical results are much better.
2.  **Gap Between Linear Theory and Deep RL:** The theory is confined to linear MDPs, which assumes a good feature representation is already available [lines: 1734, 1794]. The paper does not address the (very hard) problem of how to *learn* these features in a replicable way [lines: 1797, 1956]. The deep RL experiment, while promising, is a heuristic (quantizing Q-values, not weights [lines: 1869, 1870]) and does not have the same theoretical guarantees.
3.  **Core Set Assumption:** The generative model algorithm (R-LSVI) relies on access to a core set of state-action pairs [lines: 1701, 1704]. This is a standard assumption in related literature but is a strong one that sidesteps the exploration problem, which is (fairly) handled separately in the R-LSVI-UCB algorithm.

**Questions:**

1.  The sample complexity bounds are the main limitation. The authors mention concurrent work (Hopkins et al., 2025) achieving low overhead for replicability in the *tabular* setting [lines: 1959, 1960]. What are the primary theoretical bottlenecks that prevent a similar low-overhead result in the linear setting? Is the high cost mainly from the R-Ridge-Regression, the R-UC-Cov-Estimation, or the complex interaction between the UCB bonus and the rounding error?
2.  In the deep RL experiments, the authors chose to quantize the final Q-values [lines: 1869, 1870], whereas the theory for the linear setting involves rounding the *weights* (via R-Hypergrid-Rounding on the output of the regressor) [lines: 1516, 1443]. Could you elaborate on this design choice? Was rounding the neural network weights directly attempted, and if so, what challenges (e.g., training instability) did it present?
3.  Figure 2 shows that for PQN, regularization alone did not significantly improve policy agreement, but quantization did [lines: 1912, 1916]. This is a very interesting result. Does this imply that the instability in deep RL is less about agents converging to different *local minima* (which regularization might help smooth) and more about the high sensitivity of continuous Q-value representations to small variations in data?

---

> ### Author Response · Authors · 2025-11-20
>
> Dear Reviewer 2UsG,
> thank you for your feedback and your strong confidence in the novelty, significance, and comprehensiveness of our results.
>
> **Conceptual difference to tabular RL approach by Hopkins et al**
> The gap in performance in the tabular case stems from the fundamentally different analysis and algorithms. Our algorithm relies on a replicable exploration procedure (which is closer to Eaton et al than to Hopkins et al) that ensures that policies are identical at every stage of the algorithm. If one can move the cost of achieving replicability towards the end of the procedure, one can obtain better results. This idea laid the foundation for work such as that of the ( unpublished and concurrent work of) Hopkins et al. Intuitively, rather than achieving replicability at every step of the algorithm, they first run various non-replicable procedures and combine them in a final step at the end to obtain replicability.
>
> Following the train of thought that the results by Hopkins et al exist, and that our analysis more closely aligns with the approach by Eaton et al, one might hope that better bounds for the linear setting are achievable with a similar trick of pushing replicability down the line. However, it is not immediately clear how one might extend the result by Hopkins et al to the linear setting because it leverages the fact that there are finitely many independent states in tabular settings. Even if one could extend those results, it seems likely that a procedure like our R-Ridge regression would still be needed. Thus, we believe that our work is a first step towards understanding the problem of replicable linear RL better, which will allow us to generate new results in the future.
>
> **Rounding weights**
> We did attempt rounding weights in the first place. There are two key issues here. First, we have no control over the encoder part of the network with our rounding results. If we were to round over all weights in the encoder, we would inherit a very bad sample complexity dependence that depends on the number of parameters which is incredibly large.
> Second, learning with quantized weights in general is more difficult than learning with continuous weights simply from an optimization perspective. Thus, we chose the route that lets us optimize easily and effectively lets us mirror existing training that we know works in practice.
>
> **L2 Regularization in deep RL**
> We agree that this is a very interesting observation. Our interpretation is that several sources of instability coexist, and our experiments highlight one that standard regularization does not directly address. In many states, multiple actions receive very similar Q-values, so small run-to-run variations in the learned function can change which action attains the maximum, even when the underlying value functions are qualitatively similar. Conventional regularization (e.g., weight decay) primarily constrains the parameters and smooths the function, but it does not change the resolution of the Q-values and therefore does not remove this argmax sensitivity.
>
> We report in the Appendix that rounding leads to less churn in the phases after exploration is high and thus, quantization yields more user-controlled exploration. This does not necessarily mean that we would explore better. We suspect that it is environment-dependent on what approaches help avoid different local minima. For example, it is also not clear how this might extend to continuous action spaces where the local minimum view might be a little more involved.

---

### Official Review · Reviewer_Byb9 · 2025-10-31

**Soundness:** 4
**Presentation:** 3
**Contribution:** 3
**Rating:** 4
**Confidence:** 4

**Summary:**

This paper studies replicability in RL in the setting of linear function approximation, building upon the seminal work of Impagliazzo et al (2022) and other works that explored replicability in bandits and tabular RL. In a nutshell, replicability asks for RL algorithms that output the same policy when executed twice in the same MDP environment, under shared internal randomness. This work provides the first replicable algorithms for RL in the linear function approximation setting. En route to that task the authors first develop two new, efficient, and replicable algorithms for replicable random design regression and replicable uncentered covariance estimation. They also provide empirical evaluation of their results.

**Strengths:**

- This paper builds upon a line of work that has received interest and studies a natural problem.

- Some first theoretical bounds are established in this setting, potentially inspiring future work.

**Weaknesses:**

- I found the writing (especially in section 3) to be a bit confusing.

- The results in section 3 seem to rely on fairly standard algorithmic templates of estimating quantities from samples at some sufficiently high accuracy and then rounding; the bounds also seem to be (fairly) suboptimal.

- The situation is similar for the results in section 4; the replicability arguments are fairly standard (given the results of the previous section) and the sample complexity bounds are unsatisfactory.

**Questions:**

- I'm a bit confused about the discussion around Theorem 3.1, and some notation in it. What does $E_{D_{[t]}^M}[(\theta^\top x - y)^2 + \lambda ||\theta||^2]$ mean? In particular, since $D_{[t]}$ is a sequence of distributions, which one of them are $(x,y)$ drawn from? I might be missing something obvious. Also, I didn't understand why the discussion around the algorithm is needed. Since one can show that the estimate $\hat w$ from the vanilla ridge-regression is close to the optimal value (as stated in the theorem), why doesn't replicability follow from the (relatively) standard techniques of saying that across two executions $\hat w_1, \hat w_2$ will be close to each other (since they are close to $\theta^*$), and then apply the rounding?

- Why does Algorithm 4 need to have access to a coreset? Is that part of the standard description of RL with linear function approximation in the non-replicable setting?

- As a general comment, I would appreciate it if you could help me understand the results and the necessity of the ideas and techniques in section 3 better. I'm happy to adjust my score.

---

> ### Author Response · Authors · 2025-11-20
>
> Dear Reviewer Byb9,  thank you for your feedback and for highlighting the soundness of our work.
>
> **Replicability proofs.**
> We believe it may be a misconception that our techniques follow a standard template. To highlight the novelty of all our results, we provided a detailed explanation in the common response. For replicability of ridge regression and our main R-LSVI-UCB result we note the following:
>
> Opposite to many other results, our ridge regression results do *not* rely on high accuracy and simply rounding. Instead, our goal is to construct a *low-loss* solution that is close to a hypothetical ridge minimizer, and to use this minimizer as an anchor. This perspective is crucial for understanding why we can employ the *gradient* uniform convergence analysis.
>
> Similarly, the replicability analysis for our main result, Theorem 4.2, is non-standard. Typical arguments assume non-adaptive distributions, whereas our distribution changes as the algorithm explores, creating dependence between past and current rounds. Our key novel observation is that we can condition on the entire history of policies. The policy chosen in round t−1 determines the distribution of trajectories in round t, so the data at round t is a priori dependent on all earlier internal randomness. By conditioning on the event that both executions produce identical sequences of policies up to round t, we “factor out’’ this dependence. Once the policy is fixed, the data is a draw from a distribution that no longer depends on the past, letting us apply our primitive replicability guarantees round-by-round, even though the policies (and thus future data distributions) are adaptive. We combine this conditional argument with a strong induction over rounds.
>
> **Distribution $D_{[t]}^M$**
> Thank you for the pointer, this is a missing reference that was not properly propagated from the Appendix when we generalized the result to multiple distributions. We call $S \sim D_{[t]}^{M}$ a sample generated by taking $M$ i.i.d. draws from each of the t distributions in $D_{[t]}$. We have fixed this in the paper.
>
> **R-Ridge replicability**
> The idea that the review outlines demonstrates a correct understanding of what our approach does. The question is how to prove parameter closeness and what rates we obtain.
>
> For Theorem 3.1, the most straightforward analysis would first obtain an excess risk bound and then turn this into parameter closeness. Concretely, one can use standard Rademacher uniform convergence to show that $R(\theta) - R(\theta^* )$ is bounded and then deduce that $||\theta - \theta^* ||$ is small by strong convexity. This analysis goes through, but in the replicable setting it leads to suboptimal rates because we need $||\theta - \theta^* ||$ to be *very* small and the upper bound on the parameter distance can be something like $||\theta - \theta^* ||  \leq O(\sqrt{\varepsilon’ / \lambda})$
>
> We use a different route that is new in the replicability literature. Instead of controlling $|| \theta - \theta^* ||$ via scalar excess risk, we first prove a gradient uniform convergence bound on $\sup_{\theta} |\nabla R(\theta) - \nabla \hat{R}(\theta)|$, which is a stronger condition. This requires dealing with the vector-valued object $\nabla R(\theta)$, which rules out a vanilla scalar Rademacher analysis. We use a vector-valued symmetrization result, and a corresponding gradient uniform convergence result. These tools are not novel to our manuscript. Yet, they have not been used in the context of replicability or to obtain results of our form. We directly extend the previous results to a setting in which data is drawn independently but not necessarily from the same distribution to match the RL setting. This is a novel extension of the existing results.
>
> Building on this, our Theorem A.2 gives the application of such a gradient uniform convergence bound to ridge regression, and Lemma A.2 derives the bound on the parameter difference of convex minimizers under gradient perturbations. This gives us the tighter parameter distance $||\theta - \theta^* || \leq O(\varepsilon’ / \lambda)$. Appendix A.3 shows how these results, combined with randomized rounding, yield replicable regression.
>
> **Core-set**
> TLDR: Yes, this is a standard assumption in RL with linear functions, but our version is slightly weaker than the usual invertible empirical covariance assumption.
>
> In more detail: In general, any state can have a large impact on optimality. Consider a tabular MDP with all 0 rewards except for a single state-action pair: if we do not see this pair, we cannot find an optimal solution. Here, one typically assumes that one can sample data for each state-action pair. In the linear case, there can be infinitely many state-action pairs related by the feature transformation $\phi$. Instead of assuming access to data from all state-action pairs, standard RL work assumes access to a set of vectors in the space of $\phi$ that covers the low-dimensional space well.

---

### Official Review · Reviewer_aY1c · 2025-11-01

**Soundness:** 3
**Presentation:** 2
**Contribution:** 2
**Rating:** 4
**Confidence:** 3

**Summary:**

The paper studies establishing replicability of RL algorithm. By replicability, it means that the algorithm has the same output on arbitrary input with high probability. The authors first derive a replicability result on ridge regression and covariance estimation, which extends the result of randomized rounding result from Impagliazzo et al., 2022. Then, these tools are utilized for the analysis of replicability result of offline RL and online with exploration in the linear MDP case, which requires ridge regression and covariance estimation steps.

**Strengths:**

1. The authors tackle an important issue of reproducibility of RL algorithms in theoretical sense. This is important as many deep RL algorithms are not easy to reproduce in practice.

2. This seems to be the first result for replicability in linear MDP setting. The authors manage to connect the replicability result in supervised learning with theoretical RL in linear MDP literature.

3. The theoretical results are complemented by experimental results to show that there are room for  improvement in the sample complexity result.

**Weaknesses:**

1. While the theoretical results are new, the novelty of the algorithm or the proof is not clearly explained : 1) Section 3.1 does not clearly explain where the novelty of the algorithm lies compared with Esfandiari et al., 2023a; 2)The use of a core-set appears closely aligned with ideas already established in the literature (Lattimore and Szepesvári).

2. The overall bound is not tight compared to that of Esfandiari et al., 2023a or Hopkins et al., 2025. The authors argue that this comes at the expense of generality. However, the rationale behind this tradeoff is not entirely convincing.

3. Algorithm 2 requires additional projection  of the estimated covariance matrix on to the cone after rounding is executed.

**Questions:**

1.  The author could provide more comments on why the dependency on $\epsilon$ is high in Theorem 4.1 or 4.2.


2. As mentioned in the weakness, where does the large gap compared to the tabular case(Hopkins et al., 2025) come from?


3. Can the authors provide more detailed comparison on how worse the bound is compaed to those of tabular cases?

---

> ### Author Response · Authors · 2025-11-20
>
> Dear Reviewer aY1c,
> Thank you for your feedback and for acknowledging the importance of our work.
>
> **Novelty of our results**
> The majority of our analysis and proofs is not taken from prior work. While we rely on some prior techniques to obtain the rates we get, the main results in our paper are all novel and the analyses do not trivially follow from what is known. To highlight this, we have provided a more detailed outline of the novelty of our work in the common response.
>
> **Tightness of our bounds and sample complexity**
> Whether or not our bounds are tight or not is unclear since we do not know of a lower bound in the setting we consider. Both Esfandiari et al., as well as Hopkins et al. consider less general settings than us. Linear MDPs are strictly more complex than bandits or tabular MDPs. We would also like to point out that the original bounds in the tabular setting by Eaton et al were also not tight. However, their work inspired follow-up work that provided better rates.
>
> Regarding the dependence in Theorems 4.1 and 4.2. We agree that the dependence on the feature dimension is quite large and we are happy to clarify where it comes from. The bounds are inherited from our base replicable regression and covariance estimators, and then amplified by the way RL repeatedly “estimates from estimates.” In our setting, each Bellman backup uses targets that already depend on previous regression and covariance estimates, and we need all of these intermediate quantities to be accurate and aligned across two executions to guarantee replicability. This leads to union bounds across stages, core-set elements, and exploration rounds on top of the already non-trivial dimensional dependence of the base primitives. Our focus in this work is to establish the existence of replicable algorithms for linear RL with finite-sample guarantees rather than to optimize all exponents. Tightening these rates is an interesting direction for future work.
>
> Moreover, there may have been a misunderstanding around some of the wording in the text. The statement about ‘’expense of generality” stems from the fact that we need a more general regression statement for our RL results than what the bandits paper provides. Yet, the sample complexity of our statement almost reduces to the results provided by Esfandiari et al.. This seems like a property one would hope for in a generalized statement.
>
> **Projection of covariance**
> There may be some confusion about this step and we clarified this in the text. This step is simply part of the algorithm, and it comes at no sample-complexity overhead.
>
> **Conceptual difference to tabular approach by Hopkins et al**
> The gap in performance in the tabular case stems from the fundamentally different analysis and algorithms. Our algorithm relies on a replicable exploration procedure (which is closer to Eaton et al than to Hopkins et al) that ensures that policies are identical at every stage of the algorithm. If one can move the cost of achieving replicability towards the end of the procedure, one can obtain better results. This idea laid the foundation for work such as that of the (unpublished and concurrent work of) Hopkins et al. Intuitively, rather than achieving replicability at every step of the algorithm, they first run various non-replicable procedures and combine them in a final step at the end to obtain replicability.
>
> Following the train of thought that the results by Hopkins et al exist, and that our analysis more closely aligns with the approach by Eaton et al, one might hope that better bounds for the linear setting are achievable with a similar trick of pushing replicability down the line. However, it is not immediately clear how one might extend the result by Hopkins et al to the linear setting because it leverages the fact that there are finitely many independent states in tabular settings. Even if one could extend those results, it seems likely that a procedure like our R-Ridge regression would still be needed. Thus, we believe that our work is a first step towards understanding the problem of replicable linear RL better, which will allow us to generate new results in the future.
>
> **Tabular Rates**
> Ignoring horizon factors, the traditional, non-replicable bounds in tabular settings have been brought down to $\tilde{O}(|S| |A|/ (\varepsilon^2 ))$. The original results on replicability by Eaton et al were on the order $\tilde{O}(|S|^7 |A|^7 / (\varepsilon^5 \rho^5 ))$, which is quite impractical. Recently, Hopkins et al. provided a concurrent result to ours in the tabular setting. They establish an algorithm for which they claim a sample complexity of $\tilde{O}(|S|^3 |A| / (\varepsilon^2 \rho^2 ))$. The results in our paper deal with the significantly more general setting of linear MDPs, which makes it difficult to compare our bounds to the ones mentioned here.

---

### Official Review · Reviewer_jZdx · 2025-11-01

**Soundness:** 3
**Presentation:** 2
**Contribution:** 2
**Rating:** 4
**Confidence:** 3

**Summary:**

This paper develops new replicable algorithms for reinforcement learning in the linear function approximation regime. There have been two prior works on replicable reinforcement learning and one concurrent work. Prior works operate in the setting where the number of actions and state spaces are bounded (and small). In contrast, the present work handles the case where the number of state and action spaces are infinite. However, it makes a linear feature representation of state-action pairs, which is also widely studied in reinforcement learning. In this model, they develop a replicable algorithm for reinforcement learning. This algorithm relies on two fundamental replicable primitives: (1) ridge regression algorithm and (2) uncentered covariance estimation algorithm. The covariance estimation algorithm appears to be novel, while the ridge regression algorithm is not completely novel: it existed for fixed design settings, and this paper extends it to the i.i.d. design setting when the distribution of covariates is bounded. Finally, the authors evaluate their algorithms experimentally and show that they lead to more consistent policies.

**Strengths:**

Replicability is an interesting and valuable property of algorithms worth having. The paper presents a new replicable algorithm that operates the regime with a large number of state and action pairs where existing methods do not work. As such, I believe it's well in scope for ICLR, and studies an interesting topic.

**Weaknesses:**

I found this paper very difficult to read.

At the highest level, there are two main issues. First, the novelty of the proof ideas compared to prior work in replicability and replicable RL is unclear. Second, the paper lacks comparison with existing RL algorithms in the linear regime—it's unclear whether this algorithm modifies existing approaches by using replicable subroutines or requires more fundamental changes.

At a lower level, the very large polynomial factors in the RL results are puzzling. Can these be improved, or are they necessary? While the simulations are a nice addition. However, then I am confused why the computation of the specific polynomial factors is necessary/important: if the subroutines are replicable, shouldn't the RL algorithm automatically be replicable? Why compute the overall sample complexity? I would have felt better about the large polynomial factors if there were concrete reasons why the polynomial dependences could not be improved (easily). A proof overview here would have been nice, but as far as I see, there were no proof overviews in the main body at least.

Regarding readability issues:
The discussion before Theorem 3.1 is a bit confusing. I understand that the objective is strongly convex and hence the closeness in function values is proportional to the closeness in parameters space. But then the discussion of what is needed in the proof is not clear.

**Questions:**

Minor issues with Theorem 3.1: The phrase "from which we draw $M$ independent samples totalling $N$" is unclear. Perhaps it meant "We draw $M$ independent samples from each distribution, totaling $N = M * t$." In the definition of $\theta^*$, which distribution is the expectation taken over?

---

Minor Typos/Grammatical errors
1. Algorithm 2: “paramter”
1. Malahanobis → Mahalanobis
1. Related works: intrduced
1. Section 5.2: unforseen
1. Below theorem 3.1: staightforward
1. Above Section 5:
    1. inpractice → in practice
    1. meticuluous → meticulous
1. Section 5.1 end of first para has a repeated “the”
1. Above Section 6: Tegularization → Regularization
1. Punctuation/hyphenation missing at various places, e.g., in paragraph 2 of the paper “algorithms internal randomness” should be “algorithm’s internal randomness”
1. Section F: house-hold grade → household-grade
1. Section G: “sentence of word” → “sentence or word” also, in this section, I do not understand what “and work to specific theorem statements.” means.
1. Many of the proofs throughout the appendix use $E$ instead of $\mathbb{E}$ for denoting expectation.

Further minor suggestions
1. The “i.e.” before Proposition 2.1 is unusual.
1. Spurious space at the period at the end of proposition 2.1.
1. $\theta^*$ is mentioned in the statement of Theorem 3.2 but only defined in the statement of Theorem 3.1

---

> ### Author Response · Authors · 2025-11-20
>
> Dear Reviewer jzDx,
> Thank you for your feedback and highlighting the value of our contribution.
>
> **Proof overview**
> We appreciate the feedback. We originally tried to give high level pointers on how each proof works at the end of each theorem section. We understand that these are rather brief and have updated our revised manuscript with significantly more detail closer to our summary in the common response.
>
> **Novelty of our results**
> The majority of our analysis and proofs is not taken from prior work. While we rely on some prior techniques to obtain the rates we get, the main results in our paper are all novel and the analyses do not trivially follow from what is known. The novelty of our results is also outlined in more detail in the common response.
>
> **On downstream replicability**
> The replicability of our downstream algorithms requires its own analysis and does not immediately follow from the replicability of the primitives. Especially the analysis for the exploration algorithm is rather involved. The sub-routines are all situated in settings where there are no dependencies between data points and distributions. The latter can change in RL. Thus, it would not be immediately clear why replicability should follow. We derive various new results that demonstrate why using our routines in an RL setting is doable (see summary in general comment and Appendix D).
>
> **Description of R-Ridge proof**
> We appreciate the pointer. We have provided a more elaborate explanation in the common response and updated the manuscript accordingly. We will explicitly provide a copy of the ridge part here:
>
> For Theorem 3.1, the most straightforward analysis would be to first obtain an excess risk bound and then turn this into parameter closeness. Concretely, one can use a standard Rademacher-based uniform convergence argument to show that $R(\theta) - R(\theta^* )$ is bounded and then deduce that $|| \theta - \theta^* ||$ is small by strong convexity. This analysis goes through, but in the replicable setting, it leads to suboptimal rates because we need $|| \theta - \theta^* ||$ to be *very* small.
>
> We therefore use a different route that is, to our knowledge, new in the replicability literature. Instead of directly controlling $|| \theta - \theta^* ||$ via scalar excess risk, we first prove a gradient uniform convergence bound of the form $\sup_{\theta} | \nabla R(\theta) - \nabla \hat{R}(\theta) |$ which is a strictly stronger condition than standard risk uniform convergence. This requires dealing with the vector-valued object $\nabla R(\theta)$, which rules out a vanilla scalar Rademacher analysis. Instead, we use a vector-valued symmetrization result and a corresponding gradient uniform convergence result. These two tools are not novel to our manuscript. However, they have not been used in the context of replicability or to obtain results of our form. We directly extend the previous results to a setting in which data is drawn independently but not necessarily from the same distribution to match the RL setting. This is a novel extension of the existing results.
>
> Building on this, our Theorem A.2 gives the explicit application of such a gradient uniform convergence bound to ridge regression, and our Lemma A.2 derives a our bound on the parameter difference of convex minimizers under gradient perturbations. Appendix A.3 then shows how these results, combined with randomized rounding, yield replicable regression.
>
> **Comparison to existing linear work**
> Would you be willing to clarify what you mean by this point? There are no other replicable RL algorithms in the linear regime. Are you referring to a statement about existing known bounds inside the manuscript for non-replicable algorithms? If so, the original LSVI-UCB algorithm has a dependency of $\tilde{O}(d^3 H^4 / \varepsilon^2 )$.
>
> **Distribution $D_{[t]}^M$**
> Thank you for the pointer, this is a missing reference that was not properly propagated from the Appendix when we generalized the result to multiple distributions. We call $S \sim D_{[t]}^{M}$ a sample generated by taking $M$ i.i.d. draws from each of the t distributions in $D_{[t]}$. We have fixed this in the paper.

---

> ### Author Response · Authors · 2025-11-20
>
> **Sample complexity of RL algorithms**
> Regarding the dependence in Theorems 4.1 and 4.2. We agree that the dependence on the feature dimension is quite large and we are happy to clarify where it comes from. The bounds are inherited from our base replicable regression and covariance estimators, and then amplified by the way RL repeatedly “estimates from estimates.” In our setting, each Bellman backup uses targets that already depend on previous regression and covariance estimates, and we need all of these intermediate quantities to be accurate and aligned across two executions to guarantee replicability. This leads to union bounds across stages, core-set elements, and exploration rounds on top of the already non-trivial dimensional dependence of the base primitives. In other words, we need to iteratively apply these primitives with very high accuracy.
>
> **On improving bounds (edited to add more context)**
> At present, we do not know how to get better results given the approach we chose. It is possible that a different approach that does not leverage replicable per round exploration might achieve better results as demonstrated in the tabular setting by the concurrent work of Hopkins et al. At present, there exists no lower bound in the linear RL setting. Our focus in this work is to establish the existence of replicable algorithms for linear RL with finite-sample guarantees. Tightening the rates is an interesting direction for future work
>
> The gap in performance in the tabular case stems from the fundamentally different analysis and algorithms. Our algorithm relies on a replicable exploration procedure (which is closer to Eaton et al than to Hopkins et al) that ensures that policies are identical at every stage of the algorithm. If one can move the cost of achieving replicability towards the end of the procedure, one can obtain better results. This idea laid the foundation for work such as that of the (unpublished and concurrent work of) Hopkins et al. Intuitively, rather than achieving replicability at every step of the algorithm, they first run various non-replicable procedures and combine them in a final step at the end to obtain replicability.
>
> Following the train of thought that the results by Hopkins et al exist, and that our analysis more closely aligns with the approach by Eaton et al, one might hope that better bounds for the linear setting are achievable with a similar trick of pushing replicability down the line. However, it is not immediately clear how one might extend the result by Hopkins et al to the linear setting because it leverages the fact that there are finitely many independent states in tabular settings. Even if one could extend those results, it seems likely that a procedure like our R-Ridge regression would still be needed. Thus, we believe that our work is a first step towards understanding the problem of replicable linear RL better, which will allow us to generate new results in the future.

---

> > ### Comment · Reviewer_jZdx · 2025-11-28
> >
> > Thank you for the detailed response. I’ve read your clarifications and have no further questions.
> > However, since there are significant changes in the main body, this most likely requires an additional round of reviews from scratch.
> > Due to this, I will maintain my current score and hope the authors can incorporate the discussions in the next iteration of their manuscript.

---

> > > ### Author Response · Authors · 2025-11-29
> > >
> > > None of our hypothesis, algorithms or claims have changed. We used the feature that all authors get access to and is part of the ICLR reviewing process to address concerns and questions the reviewers requested elaboration on. We simply added more description to our proofs as requested by multiple reviewers.

---

### Author Response · Authors · 2025-11-20
**Common Response**

Dear Reviewers,

thank you all for your feedback on our submission. The feedback we received was very clear. It seems there is consensus on how the current version of the manuscript can be improved, which is exactly what one might hope for from a set of scientific reviews. In general, it seems that reviewers had trouble following the proof structure in section 3 and thus the contribution inherent to our analysis. To address this, we have updated our manuscript as follows

* We acknowledge that we were a bit ambitious in attempting to provide all results, algorithms, and figures in the main text, which led to our proof descriptions ending up a bit short due to space constraints. We have spent a large chunk of the newly available additional page of content towards elaborating on our findings and the novelty of our contribution.
* We increased the number of evaluation trajectories in our experimental results on Atari. Now, increased agreement also correlates with a decrease in return variance, as one might expect. We updated both plots; none of the claims changed.

In the updated manuscript, we highlight all major changes from the previous version in blue. We will summarize the major changes here and provide point-wise answers to each reviewer in the respective review sections.

---

> ### Author Response · Authors · 2025-11-20
>
> While many of our results rely on randomized rounding in some form or another, the analyses of our algorithms are non-traditional and novel; i.e. one cannot simply apply rounding and hope to get away with it.
>
> **Ridge Regression**
> What exists: Esfandiari et al. provide an algorithm for replicable least-squares estimation when the covariates are fixed, can be sampled freely, and form a full-rank basis. The algorithm samples each covariate a certain number of times, rounds the noisy label to a consistent one, and then fits a regressor. None of the requirements for this algorithm are met in the RL setting with exploration. Data points can be arbitrary vectors in the space of $\phi$, we rely on exploring the space and cannot sample freely and initially, we may not have enough data to cover the full space. Our R-Ridge algorithm is the first replicable algorithm for regression that meets all the requirements.
>
> We cannot rely on label rounding because when the covariates are not fixed, it is not clear which point to cluster to the same label. Thus, we would like to round weights instead. However, the weights of an MLE estimator are not guaranteed to be close and we leverage the convex objective of ridge regression instead.
>
> For Theorem 3.1, the most straightforward analysis would be to first obtain an excess risk bound and then turn this into parameter closeness. Concretely, one can use a standard Rademacher-based uniform convergence argument to show that $R(\theta) - R(\theta^* )$ is bounded and then deduce that $|| \theta - \theta^* ||$ is small by strong convexity. This analysis goes through, but in the replicable setting, it leads to suboptimal rates because we need $|| \theta - \theta^* ||$ to be very small.
>
> We therefore use a different route that is, to our knowledge, new in the replicability literature. Instead of directly controlling
> $|| \theta - \theta^* ||$ via scalar excess risk, we first prove a gradient uniform convergence bound on $\sup_{\theta} \| \nabla R(\theta) - \hat{R}(\theta) \|$ which is a strictly stronger condition than standard risk uniform convergence. This requires dealing with the vector-valued object $\nabla R(\theta)$ which rules out a vanilla scalar Rademacher analysis. Instead, we use a vector-valued symmetrization result and a corresponding gradient uniform convergence result. These tools are not novel to our manuscript. However, they have not been used in the context of replicability or to obtain results of our form. We directly extend the previous results to a setting in which data is drawn independently but not necessarily from the same distribution to match the RL setting. This is a novel extension of the existing results.
>
> Building on this, our Theorem A.2 gives the explicit application of such a gradient-uniform-convergence bound to ridge regression, and our Lemma A.2 derives our bound on the parameter difference of convex minimizers under gradient perturbations. Appendix A.3 then shows how these results, combined with randomized rounding, yield replicable regression.
>
> *Core Set*
> We leverage a core-set assumption like other works. However, our core set differs in that it is not defined via an invertible covariance matrix but via a weaker geometric coverage condition. This is an assumption not commonly used in the literature and it implies that the standard least-squares analysis does not apply directly because the covariance cannot necessarily be inverted.
>
> For Theorem 3.2, this allows us to state a version of ridge regression in which we get error guarantees despite the lack of full rank. The guarantee follows from triangle inequality by demonstrating that the ridge minimizer is not too far from the true minimizer under this assumption. The proof for this is specific to our assumptions and is provided in Appendix A.4.
>
> **Covariance estimation**
> To the best of our knowledge, we provide the first algorithm for replicable uncentered covariance estimation.
>
> For Theorem 3.3, a first thought one might have is that one can simply compute a covariance matrix, apply concentration elementwise and apply the matrix rounding technique from Impagliazzo et al.. However, this would not yield a valid covariance matrix. Since one rounds each entry to its own grid interval, the top and bottom triangular parts of the matrix would not be identical and symmetry would be broken. We resolve this by only partially rounding the matrix and then explicitly copying it. Then, rounding can flip small eigenvalues of the matrix to be negative. The resulting matrix would not be positive semi-definite. To resolve this, we additionally clip the eigenvalues and return the projected matrix. A key observation is that this step can not move us further away from the true covariance, which means it does not incur any additional sample complexity. While the overall procedure can be summarized succinctly, implementing it to simultaneously preserve symmetry, PSD, and replicability requires careful design.

---

> > ### Author Response · Authors · 2025-11-20
> >
> > **R-LSVI (core set)**
> > The analysis of Theorem 4.1 is arguably the most straightforward across all our results. It combines the core-set analysis from Lattimore et al. adapted to our weaker core set and to ridge regression, and an induction similar to that of Eaton et al. While mechanically simple, the result is not an immediate corollary and requires its own proof. What follows is the first algorithm for replicable RL with linear function approximation. We find it instructive to include as it demonstrates the cost of planning when no exploration is needed.
> >
> > **R-LSVI-UCB**
> > This is the main result of our work. Again, there exists no replicable RL algorithm with linear function approximation in the exploration setting and the algorithm we provide is novel.
> >
> > One might be tempted to simply take the ridge and covariance results and apply them to a standard linear RL algorithm. This breaks in multiple ways and requires very careful consideration.
> >
> > *Batching:*
> > A natural first idea is to take the standard LSVI-UCB algorithm of Jin et al. (2020) and simply replace each least-squares / covariance computation with our primitives. However, LSVI-UCB updates the value estimates after every episode using all past data in a fully online way. The analyses of R-Ridge and R-UC-Cov, on the other hand, require that each call is run on a batch of samples of a fixed size, and the concentration bounds are proved for that batch size. If we tried to call the replicable primitives in the fully online LSVI-UCB loop, we would no longer be in the regime covered by our uniform convergence and replicability guarantees, and we could not control the total rounding error across all updates.
> >
> > This can be addressed by batching trajectories. This batching procedure, however, forces us to break an algorithmic symmetry that exists in the LSVI-UCB algorithm and complicates our analysis. The original LSVI-UCB algorithm computes both the UCB bonus term and linear regression at every iteration using the same Gram matrix: the one produced by all data collected by the algorithm thus far. However, to ensure useful concentration (which we need to argue replicability) of the matrix used to compute the bonus term, we take an average over the batch of trajectories collected in every round. Thus, the linear UCB bonus term for exploration of a state-action pair is not as tightly related to the estimation error for that state-action pair as in the original algorithm.  The resulting theorem (Thm. 4.2) is therefore not a direct corollary of Jin et al. (2020), but a new analysis tailored to this batched, replicable variant.
> >
> > *Rounded UCB bonus:*
> > The optimism and regret guarantees in Jin et al. (2020) rely on the UCB bonus being computed from the regularized Gram matrix, and the proof uses an elliptical potential argument. If we simply rounded the covariance matrix to get replicability and then plugged this rounded matrix into the original proof, the classical argument no longer applies, because it assumes the potential is computed with the true Gram matrix, not with an arbitrary matrix-valued perturbation of it.
> >
> > We therefore carry out a new perturbation analysis for the UCB bonus under matrix rounding. Lemma D.2 bounds Mahalanobis norms under a PSD perturbation of the gram matrix; Lemma D.3 uses this to control the effect of the R-UC-Cov rounding step on both the rewards and the bonuses; Lemma D.4 re-establishes the UCB property for our rounded Gram matrices; and Lemma D.5 + Corollary D.1 show that the contribution of the UCB bonus to regret remains of the same order as in the unrounded LSVI-UCB analysis. None of these steps are needed in the original proof. They are specifically required to make UCB-style optimism and regret go through in the presence of the randomized rounding.
> >
> > *Identical distribution:*
> > The replicability guarantees for our primitives are proved for sequences of distributions that are fixed independently of the algorithm’s internal randomness. If we naively ran these primitives inside the standard adaptive LSVI-UCB loop, the data distribution at time t would depend on previous policies and the assumptions of Thms. 3.1 and 3.3 are violated. We no longer know that the two executions see samples from the same sequence of distributions, so simply union-bounding the primitive-level replicability guarantees would not give replicability of the overall RL algorithm.
> >
> > We prove replicability of R-LSVI-UCB by strong induction over rounds: assuming that all previous rounds have produced identical policies in the two executions, the trajectories collected in the next round are i.i.d. from the same distribution in both runs. Under this conditioning, the assumptions of Thms. 3.1 and 3.3 are satisfied, so the calls to R-Ridge and R-UC-Cov are themselves replicable. This inductive argument is specific to the exploration setting and it gives a new kind of adaptive replicability proof.

---

### Meta-Review · Area_Chair_93UR · 2026-01-05

**Summary:**

One of the main concerns is the technical novelty of the paper from Reviewer jZdx, Reviewer aY1c, Reviewer Byb9, especially the RL analyses. On the other hand, Reviewer 2UsG points out the novelty of the new core regression tool on which RL algorithms are later built on.  The reviewers also pointed out the final rate on RL seems highly suboptimal, the theory is detached from the deep RL experimental results, and some writing is hard to parse. In the rebuttal, I think the authors provide sufficient clarity on technical challenges they overcame, and where the technical novelty stems from. They also improved the readability with revision. I think the paper as a whole, combining new replicable regression algorithms suitable for subroutines for RL algorithms and the replicable RL algorithms themselves, warrant enough technical contribution. The gap between theory and deep RL, and the suboptimal rates, while not perfect, does occlude the paper's step to push the literature further. Since all reviewers commend on the logical correctness, I would proceed to recommend acceptance. I would advise the authors to take the rebuttal and further clarify on these aspects in the revision.

**Reviewer Concerns:**

See summary.

**Reviewer Scores:**

Reviewer jZdx remains their score, though I don't they fully read the revision/rebuttal.
Reviewer aY1c, Reviewer Byb9 may increase their scores a little.
Reviewer 2UsG would remain their original positive assessment.

---

### Decision · Program_Chairs · 2026-01-26

Accept (Poster)